# Cortical adaptation to sound reverberation

**Aleksandar Z Ivanov\*, Andrew J King, Ben DB Willmore\*[†], Kerry MM Walker\*[†], Nicol S Harper\*[†]**

Department of Physiology, Anatomy and Genetics, University of Oxford, Oxford, United Kingdom

**Abstract** In almost every natural environment, sounds are reflected by nearby objects, producing many delayed and distorted copies of the original sound, known as reverberation. Our brains usually cope well with reverberation, allowing us to recognize sound sources regardless of their environments. In contrast, reverberation can cause severe difficulties for speech recognition algorithms and hearing-impaired people. The present study examines how the auditory system copes with reverberation. We trained a linear model to recover a rich set of natural, anechoic sounds from their simulated reverberant counterparts. The model neurons achieved this by extending the inhibitory component of their receptive filters for more reverberant spaces, and did so in a frequency-dependent manner. These predicted effects were observed in the responses of auditory cortical neurons of ferrets in the same simulated reverberant environments. Together, these results suggest that auditory cortical neurons adapt to reverberation by adjusting their filtering properties in a manner consistent with dereverberation.

## Editor's evaluation

This study identifies a mechanism based on context-dependent plasticity of inhibitory receptive fields that likely plays a role in suppression of reverberation signals in hearing. This new mechanism is a very interesting starting point to describe the biological circuit underpinnings of reverberation suppression, a complex signal processing ability of the auditory system.

**\*For correspondence:**
aleksandar.ivanov@dpag.ox.ac. uk (AZI);
benjamin.willmore@dpag.ox.ac. uk (BDBW);
kerry.walker@dpag.ox.ac.uk (KMMW);
nicol.harper@dpag.ox.ac.uk (NSH)

[†]These authors contributed equally to this work

## Introduction

Reverberations accompany almost all natural sounds that we encounter and are the reflections of sound off objects in the environment, such as walls, furniture, trees, and the ground (*Huisman and Attenborough, 1991*; *Sakai et al., 1998*). Compared to the original sound, these reflections are attenuated and distorted due to frequency-selective absorption and delayed due to increased path length (*Kuttruff, 2017*).

Reverberation can be useful, helping us judge room size, sound-source distance, and realism (*Shinn-Cunningham, 2000*; *Trivedi et al., 2009*; *Kolarik et al., 2021*). However, strong reverberation can impair sound-source localization (*Hartmann, 1982*; *Shinn-Cunningham and Kawakyu, 2003*; *Rakerd and Hartmann, 2005*; *Shinn-Cunningham et al., 2005*) and segregation (*Culling et al., 1994*; *Darwin and Hukin, 2000*), pitch discrimination (*Sayles and Winter, 2008*), and speech recognition (*Knudsen, 1929*; *Nábělek et al., 1989*; *Guediche et al., 2014*; *Houtgast and Steeneken, 1985*). Notably, reverberation can be detrimental for people with hearing impairments, increasing tone detection thresholds and reducing intelligibility of consonants (*Humes et al., 1986*; *Helfer and Wilber, 1990*). It can also impede the effectiveness of auditory prostheses such as hearing aids (*Qin and Oxenham, 2005*;

*Poissant et al., 2006*) and substantially reduces the performance of automatic speech recognition devices (*Yoshioka et al., 2012*; *Kinoshita et al., 2016*).

The auditory system has mechanisms to help us cope with reverberation, to the extent that healthy listeners often only directly notice it when it is strong (in environments such as cathedrals). In the presence of mild-to-moderate reverberation, healthy listeners can continue to perform sound localization (*Hartmann, 1982*; *Rakerd and Hartmann, 2005*) and speech and auditory object recognition tasks (*Houtgast and Steeneken, 1985*; *Bradley, 1986*; *Darwin and Hukin, 2000*; *Culling et al., 2003*; *Nielsen and Dau, 2010*). Because it is such a ubiquitous property of natural sounds, these findings highlight the importance, for both normal and impaired hearing, of understanding how the brain copes with reverberation (*Xia et al., 2018*).

What are the neurophysiological mechanisms that support listening in reverberant environments? Previous studies have examined subcortical processes that facilitate localization of reverberant sounds (*Yin, 1994*; *Litovsky and Yin, 1998*; *Fitzpatrick et al., 1999*; *Spitzer et al., 2004*; *Tollin et al., 2004*; *Pecka et al., 2007*; *Devore et al., 2009*; *Kuwada et al., 2012*; *Kim et al., 2015*; *Brughera et al., 2021*), and how subcortical processing of synthetic periodic sounds is disrupted by reverberation (*Sayles and Winter, 2008*) and partially restored by compensatory mechanisms (*Slama and Delgutte, 2015*). Much less is known about the neural processing of speech and other complex natural sounds in the presence of reverberation. However, converging evidence from electrophysiological recordings in animals (*Rabinowitz et al., 2013*; *Moore et al., 2013*; *Mesgarani et al., 2014*) and from human EEG (*Khalighinejad et al., 2019*) and fMRI (*Kell and McDermott, 2019*) studies suggests that representations of sounds that are invariant to non-reverberant background noise emerge at the level of auditory cortex via neuronal adaptation to stimulus statistics (but see also *Lohse et al., 2020*). Auditory cortex may play a similar role in adaptation to reverberation. Indeed, speech and vocalization stimuli reconstructed from auditory cortical responses in awake ferrets more closely resemble their anechoic versions than the reverberant ones, even if the sounds were presented in reverberant environments (*Mesgarani et al., 2014*). Similar results have been found in humans using sound reconstructions from EEG measurements (*Fuglsang et al., 2017*). It remains unclear, however, whether the observed cortical invariance to reverberation can occur in the absence of top-down attention, and through what neural mechanisms this is achieved.

Here, we addressed these questions by using a model to predict what neural tuning properties would be useful for effective attenuation of reverberation (a normative 'dereverberation model'). We then test these predictions using neural recordings in the auditory cortex of anesthetized ferrets. More specifically, we made reverberant versions of natural sounds in simulated rooms of different sizes. Next, we trained a linear model to retrieve the clean anechoic sounds from their reverberant versions. Our trained model provided specific predictions for how the brain may achieve this task: with increased reverberation, neurons should adapt so that they are inhibited by sound energy further into the past, and this should occur in a sound frequency-dependent manner. We observed these predicted effects in the responses of auditory cortical neurons to natural sounds presented in simulated reverberant rooms, and show that they arise from an adaptive process. These results suggest that auditory cortical neurons may support hearing performance in reverberant spaces by temporally extending the inhibitory component of their spectrotemporal receptive fields.

## Results

### Dereverberation model kernels show reverberation-dependent inhibitory fields

We trained a simple dereverberation model to estimate the spectrotemporal structure of anechoic sounds from reverberant versions of those sounds. The anechoic sounds comprised a rich 10-min-long set of anechoic recordings of natural sound sources, including speech, textures (e.g. running water) and other environmental sounds (e.g. footsteps) (see Sound stimuli and virtual acoustic space). Reverberation in small (3.0 × 0.3 × 0.3m) and large (15 × 1.5 × 1.5m) tunnel-shaped rooms was simulated using the virtual acoustic space simulator Roomsim (*Campbell et al., 2005*; *Figure 1A*). The simulation also modelled the acoustic properties of the head and outer ear by using a ferret head-related transfer function (HRTF, *Schnupp et al., 2001*). The dimensions of the smaller room made it less reverberant (reverberation time: $RT_{10} = 130ms$, $RT_{60} = 0.78s$) than the larger room ($RT_{10} = 430ms$, $RT_{60} = 2.6s$).

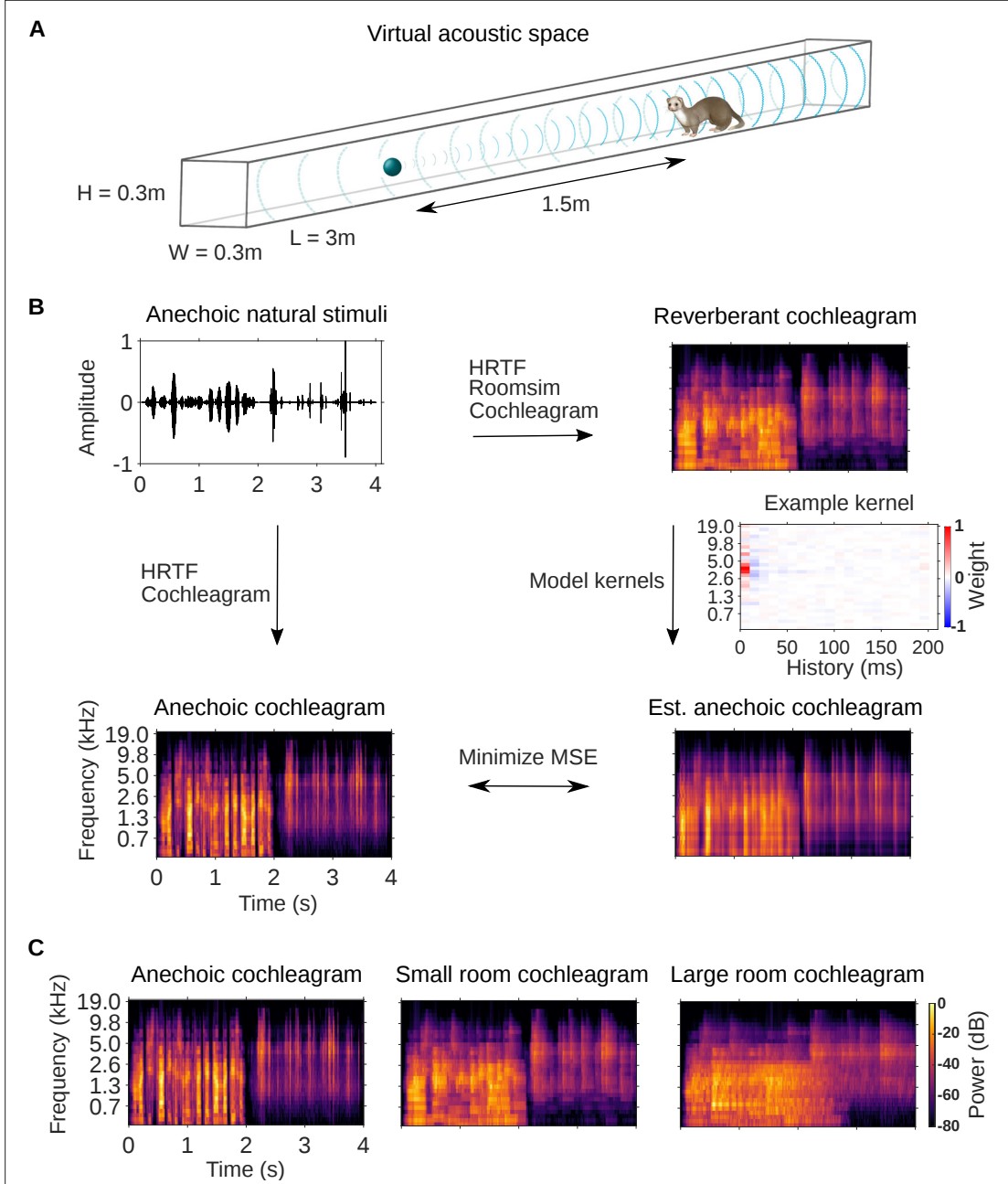

**Figure 1.** Dereverberation model. (**A**) Virtual acoustic space was used to simulate the sounds received by a ferret from a sound source in a reverberant room for diverse natural sounds. Schematic shows the simulated small room (length (L) = 3m, width (W) = 0.3m, height (H) = 0.3m) used in this study, and the position of the virtual ferret's head and the sound source (1.5m from the ferret head) within the room. We also used a medium (x2.5 size) and large room (x5). The acoustic filtering by a ferret's head and ears was simulated by a head-related transfer function (HRTF). (**B**) Schematic of the dereverberation model. The waveform (top left panel) shows a 4s clip of our anechoic recordings of natural sounds. For a given room, simulated room reverberation and ferret HRTF filtering were applied to the anechoic sound using Roomsim (**Campbell et al., 2005**), and the resulting sound was then filtered using a model cochlea to produce a reverberant cochleagram (top right panel). A cochleagram of the anechoic sound was also produced (bottom left panel). For each room, a linear model was fitted to estimate the anechoic cochleagram from the reverberant cochleagram for diverse natural sounds. Each of the 30 kernels in the model was used to estimate one frequency band of the anechoic sound. One such model kernel is shown (middle right panel). Generating the estimated anechoic cochleagram (bottom right panel) involved convolving each model kernel with the reverberant cochleagram, and the mean squared error (MSE) between this estimate and the anechoic cochleagram was minimized with respect to the weights composing the kernels. (**C**) Sample cochleagrams of a 4s sound clip for the anechoic (left panel), small room (middle panel), and large room (right panel) reverberant conditions.

After the reverberant sounds were generated, they were converted to cochleagrams (*Figure 1B*). We used a simple 'log-pow' cochlear model to produce the cochleagrams, as our recent work suggests that these cochleagrams enable better prediction of cortical responses than more detailed cochlear models (*Rahman et al., 2020*). These spectrotemporal representations of the sound approximate the cochlear filtering and resulting representation of the sound at the auditory nerve (*Brown and Cooke, 1994*; *Rahman et al., 2020*). Cochleagrams of an example sound clip presented in the anechoic, small and large room conditions are shown in *Figure 1C*.

We trained a dereverberation model to recover the anechoic cochleagram, using either the small or large room cochleagrams as an input (*Figure 1B*). The dereverberation model was comprised of a set of 'dereverberation' kernels, one for each frequency in the anechoic cochleagram (see Model kernels). Each model kernel used the full reverberant cochleagram (up to 200ms in the past) to estimate the current power in the anechoic cochleagram within a single frequency band. This resulted in a set of positive and negative weights in each model kernel. Obtaining the estimated anechoic sounds involved convolution over time between the model kernels and the reverberant cochleagrams, and the model was trained to minimize the difference between this estimate and the original anechoic sound (*Figure 1B*). The model was trained separately to dereverberate the small and large room cochleagrams. For each room, on a held-out test set, the dereverberation model reduced the difference

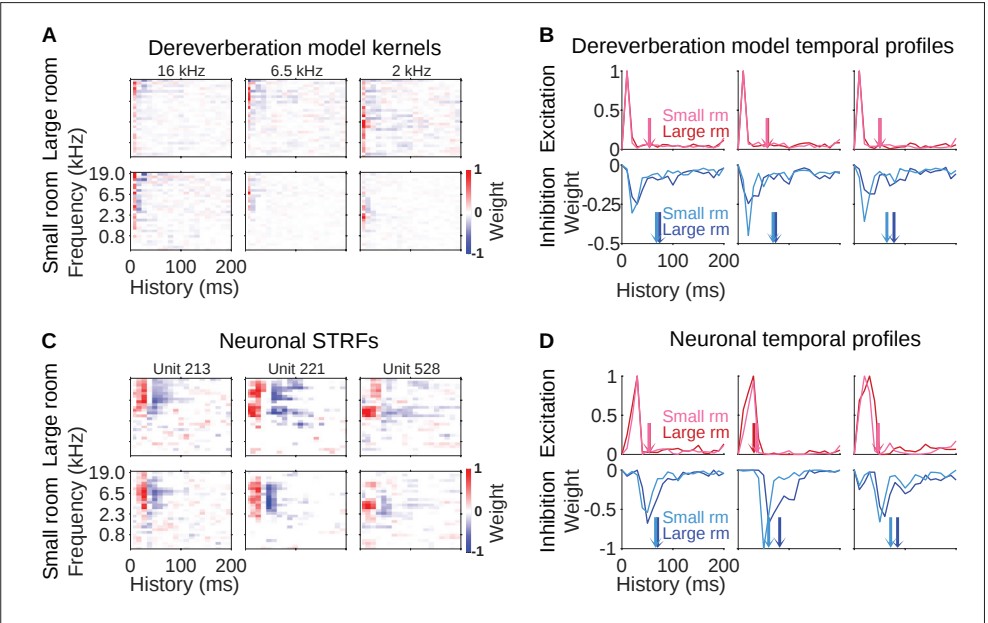

**Figure 2.** Similar reverberation effects were observed in the dereverberation model kernels and neuronal STRFs. (**A**) Example model kernels resulting from the dereverberation model. Three example model kernels are shown, after training on the large (top row) or small (bottom row) room reverberation. The frequency channel which the model kernel is trained to estimate is indicated above each kernel. The color scale represents the weights for each frequency (y-axis) and time (x-axis). Red indicates positive weights (i.e. excitation), and blue indicates negative weights (i.e. inhibition; color bar right). (**B**) Each plot in the top row shows the temporal profile of the excitatory kernel weights for the corresponding example model kernels shown in A. Excitatory temporal profiles were calculated by positively rectifying the kernel and averaging over frequency (the y-axis), and were calculated separately for the small (pink) and large (red) rooms. The center of mass of the excitation, $COM^+$, is indicated by the vertical arrows, which follow the same color scheme. The bottom row plots the inhibitory temporal profiles for the small (cyan) and large (blue) rooms. Inhibitory temporal profiles were calculated by negatively rectifying the kernel and averaging over frequency. The $COM^-$ is indicated by the colored arrows. (**C**) Spectrotemporal receptive fields (STRFs) of three example units recorded in ferret auditory cortex, measured for responses to natural sounds in the large room (top row) or small room (bottom row), plotted as for model kernels in A. (**D**) Temporal profiles of the STRFs for the three example units shown in C, plotted as for the model kernels in B.

The online version of this article includes the following figure supplement(s) for figure 2:

**Figure supplement 1.** Model kernels and neuronal STRFs across frequency channels.

**Figure supplement 2.** Model and neuronal temporal profiles across frequency channels.

between the incoming reverberant cochleagram and the anechoic cochleagram (small room mean squared error reduction 26%; large room reduction 20%).

Three examples of model kernels are shown in *Figure 2A* for the large room and the small room, with the anechoic frequency band they estimate indicated at the top. For each model kernel, the excitatory (red) and inhibitory (blue) weights represent spectrotemporal features in the reverberant cochleagrams that are associated with increased or decreased power in the specified frequency band of the anechoic cochleagram, respectively. The majority of the excitatory and inhibitory weights appear localized around a particular frequency, resembling the frequency tuning seen in auditory cortical neurons (*Bizley et al., 2005*). This is expected in our dereverberation model since each kernel aims to estimate the power in a given frequency band of the anechoic cochleagram.

The model kernels had temporally asymmetric structure, where strongest excitatory weights tended to occur first (*Figure 2A*), followed soon after by a longer inhibitory field. These excitatory and inhibitory timings are readily apparent when we plot the frequency-averaged positive and negative kernel weights (*Figure 2B*), and are a common feature across all kernels in the model (*Figure 2—figure supplement 1A*, *Figure 2—figure supplement 2A*). This pattern has been commonly observed in the spectrotemporal receptive fields (STRFs) of auditory cortical neurons (*deCharms et al., 1998*; *Linden et al., 2003*; *Harper et al., 2016*; *Rahman et al., 2019*), so our model qualitatively reproduces the basic frequency tuning and temporal characteristics of auditory cortical neurons.

Importantly, we can compare the model kernels for the large room with those for the small room. The inhibitory components of the large-room kernels tended to be delayed and longer in duration, relative to the small-room kernels (*Figure 2B*). In contrast, the temporal profile of the excitatory components was similar for the small and large rooms. We predicted that a comparable shift in inhibitory filtering could play a role in reverberation adaptation in auditory cortical neurons.

## Auditory cortical neurons have reverberation-dependent inhibitory fields

To test the predictions of our dereverberation model in vivo, we presented to anesthetized ferrets an 80s subset of the natural sounds in the simulated small and large reverberant rooms (see Sound stimuli and virtual acoustic space). We did this while recording the spiking activity of neurons in the auditory cortex using Neuropixels high-density extracellular microelectrodes (*Jun et al., 2017*; see Surgical procedure). Stimuli were presented as 40s blocks, in which all sounds were in the same reverberant room condition. This allowed neurons to adapt to the reverberation acoustics of the room. We recorded the responses of 2244 auditory cortical units. Of these, the 696 units (160 single units, 23%) that were responsive to the stimuli were used for further analysis (see Spike sorting).

We estimated the filtering properties of each unit by fitting a separate STRF to the neuronal responses for each reverberant condition. Neuronal STRFs are linear kernels mapping the cochleagram of the sound stimulus to the time-varying firing rate of the neuron (*Theunissen et al., 2001*). The positive regions of an STRF represent sound features whose level is positively correlated with the neuron's spike rate, providing the 'excitatory' part of the receptive field. Similarly, negative regions of the STRF indicate features whose level is negatively correlated with the cortical unit's spike rate, providing the 'inhibitory' receptive field.

Examples of typical neuronal STRFs are shown in *Figure 2C*, and these can be compared to the model kernel properties of our dereverberation model above (*Figure 2A*). As mentioned above, the model kernels show some similarity to the STRFs typically reported for auditory cortical neurons (*deCharms et al., 1998*; *Linden et al., 2003*; *Harper et al., 2016*; *Rahman et al., 2019*). Likewise, the model kernels show similarity to the STRFs we present here, including having frequency tuning, early excitatory receptive fields and delayed inhibitory receptive fields (*Figure 2C*). These consistencies between the general features of our model and neurophysiological responses validated our use of this normative approach to capture neural response properties. We next examined if the model could predict neural adaptation to different reverberant conditions.

The important prediction we observed in the model was that the inhibitory fields tended to be more delayed and of longer duration in the large-room kernels versus the small-room kernels, whereas the excitatory field remained unchanged. Strikingly, we observed the same pattern in the neuronal STRFs in *Figure 2D*. This observation also held across different frequency channels in both the model and the data (*Figure 2—figure supplement 1*, *Figure 2—figure supplement 2*).

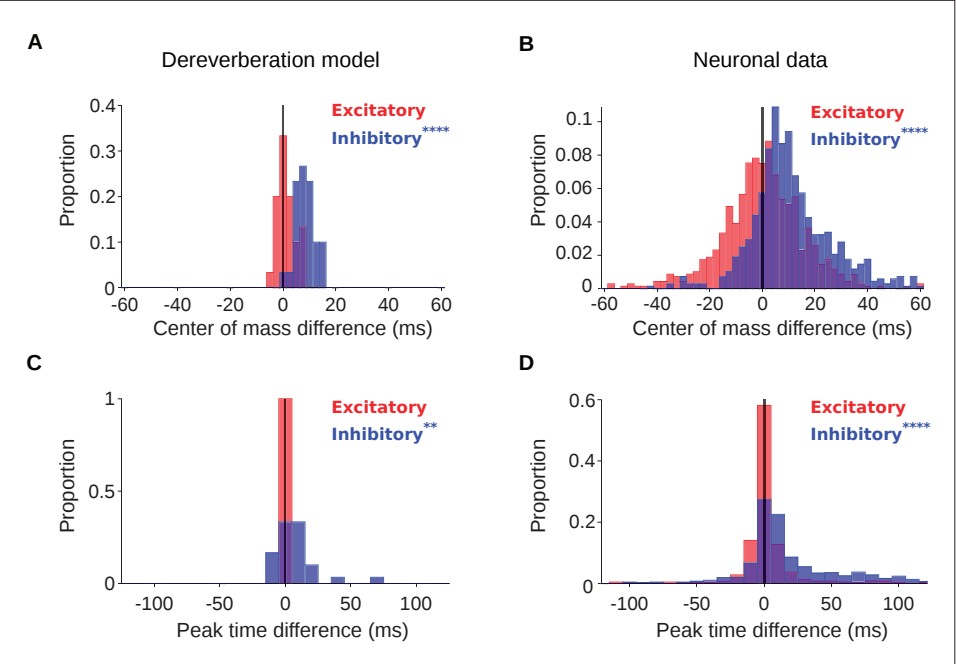

**Figure 3.** Increased reverberation produces delayed inhibitory fields in dereverberation model kernels and neuronal STRFs. (**A**) Histograms of the difference in center of mass of the temporal profiles (for the inhibitory field, $COM^-$, blue; excitatory field, $COM^+$, red) of dereverberation model kernels between the two different reverberant conditions (large - small room). The $COM^-$ were larger in the larger room, with a median difference = 7.9ms. $COM^+$ did not differ significantly between the rooms (median difference = 1.0ms). (**B**) Center of mass differences, plotted as in A, but for the auditory cortical units. The $COM^-$ increased in the larger room (median difference = 9.3ms), while $COM^+$ was not significantly different (median difference = 0.3ms). (**C**) Histograms of the large - small room difference in peak time for the temporal profiles of the model kernels (inhibitory, $PT^-$, blue; excitatory, $PT^+$, red). The $PT^-$ values were larger in the larger room (median difference = 5.3ms), whereas $PT^+$ values were not significantly different (median difference = 0.0ms). (**D**) Peak time differences for neuronal data, plotted as in C. The $PT^-$ values increased in the larger room (median difference = 9.4ms), while $PT^+$ did not significantly differ between between the two rooms (median difference = 0.0ms). Asterisks indicate the significance of Wilcoxon signed-rank tests: $^{****}p < 0.0001$, $^{**}p < 0.01$.

The online version of this article includes the following figure supplement(s) for figure 3:

**Figure supplement 1.** Analyses using the Carney Bruce Erfani Zilany (CBEZ) cochlear model.

**Figure supplement 2.** A medium room condition shows intermediate center of mass and peak time values compared to the small and large room conditions.

## Similar effects of reverberation on the inhibitory fields of model kernels and auditory cortical neurons

Since both the dereverberation model and the neuronal STRFs had structure which varied according to the reverberation condition, we sought to investigate these effects quantitatively. We used two metrics to estimate the temporal dynamics of the inhibitory (and excitatory) components of the model kernels and neuronal STRFs: Center of mass ($COM$) and peak time ($PT$) (see Quantification of the temporal effects in model kernels and neuronal STRFs). The $COM$ measured the average temporal delay of the inhibitory ($COM^-$) or excitatory ($COM^+$) components of the model kernels/neuronal STRFs (**Figure 2B and D**). The $PT$ is the time at which the maximal inhibition ($PT^-$) or excitation ($PT^+$) occurred.

For each anechoic frequency channel in the dereverberation model, we calculated the difference between the $COM^-$ for the kernels in the large room and small room conditions, providing 30 $COM^-$ differences (1 for each channel), and did the same for the $COM^+$. We plotted the distribution of these differences as histograms in **Figure 3A** (see also **Supplementary file 1** for supplementary statistics for this and other analyses). Similarly, a histogram of the $COM$ difference between the neuronal STRFs in the large and small room conditions is plotted for 696 cortical units in **Figure 3B**. We found that the

$COM^+$ did not differ significantly between the small and large rooms, either for model kernels (median $COM^+$ difference = 0.97ms, Wilcoxon signed-rank test, p = 0.066) or neuronal STRFs (median $COM^+$ difference = 0.32ms, p = 0.39). In contrast, the $COM^-$ showed clear dependence on room size. The inhibitory centers of mass were higher in the larger room for both the model kernels (median $COM^-$ difference = 7.9ms, p = $1.9 \times 10^{-6}$), and neuronal STRFs (median $COM^-$ difference = 9.3ms, p = $1.5 \times 10^{-66}$).

The results of our analysis of $PT$ were largely consistent with our $COM$ findings (**Figure 3C and D**). The peak time of the excitatory component ($PT^+$) of model kernels did not differ between the small and large room (median $PT^+$ difference = 0.0ms, p = 1.0), and neither did the $PT^+$ in the neural data (median $PT^+$ difference = 0.0ms, p = 0.38). The peak time of the inhibitory component, on the other hand, occurred later in the large room, in both the model kernels (median $PT^-$ difference = 5.3ms, p = $3.7 \times 10^{-3}$) and neuronal STRFs (median $PT^-$ difference = 9.4ms, p = $4.0 \times 10^{-44}$). We also observed these room-size-dependent delays in the $COM$ and $PT$ of inhibitory components when we used a more detailed cochlear model (**Bruce et al., 2018**; **Zilany et al., 2014**; **Zilany et al., 2009**) to generate input cochleagrams (**Figure 3—figure supplement 1**).

In general, there was more spread in the $COM$ and $PT$ in the neuronal data compared to the dereverberation model. This is likely because, unlike in the model, which was focused purely on dereverberation, the auditory cortex subserves multiple functions and a diversity of STRF spans is useful for other purposes (e.g. prediction, **Singer et al., 2018**). Despite this, it is notable that the median $COM$ and $PT$ differences of the dereverberation model were of similar magnitude to those of the neuronal data.

As our stimulus set described above included only two reverberant rooms, it was not clear if the neurons treated these simulated rooms as two points along an ordered reverberation scale. To further examine whether the timing of the neuronal STRF inhibitory component scales with the amount of reverberation in our simulated room, we added a third 'medium' sized room with the same relative proportions and absorption properties as the small and large rooms. We measured auditory cortical responses to this extended stimulus set in 2 ferrets (266 cortical units).

The $COM$ and $PT$ measures of neuronal STRF dynamics were compared across the small, medium, and large room conditions (**Figure 3—figure supplement 2**). As expected, there was little effect of room size on the timing of the excitatory STRF components (**Figure 3—figure supplement 2A,C**). The $COM^+$ showed a weak but significant overall increase with room size (Kruskal-Wallis test; $\chi^2(2) = 6.4$, p = 0.042), but there was no effect of room size on the peak time of excitation, $PT^+$ ($\chi^2(2) = 1.4$, p = 0.50). In post-hoc pairwise comparisons, $COM^+$ only differed between the small and medium rooms (Fisher's least significant differences; large-small: p = 0.21; large-medium: p = 0.21; medium-small: p = 0.012).

In contrast, and as predicted, we found that the delay of the inhibitory STRF components increased with greater room reverberation. The $COM^-$ was generally larger for larger rooms (Kruskal-Wallis test; $\chi^2(2) = 37$, p = $7.6 \times 10^{-9}$, **Figure 3—figure supplement 2B**). Post-hoc pairwise tests confirmed that $COM^-$ differed between all three reverberant conditions (Fisher's least significant differences; large-small: p = $1.3 \times 10^{-9}$; large-medium: p = $2.0 \times 10^{-4}$; medium-small: p = 0.019). The peak time of STRF inhibition, $PT^-$, also increased with room size across all three rooms ($\chi^2(2) = 27$, p = $1.6 \times 10^{-6}$; large-small: p = $2.7 \times 10^{-7}$; large-medium: p = 0.0024; medium-small: p = 0.036, **Figure 3—figure supplement 2D**).

Thus, as room size, and hence reverberation time, was increased, we observed an increase in the delay of inhibition in the tuning properties of auditory cortical neurons. This increase is consistent with a normative model of dereverberation, suggesting that the tuning properties of auditory cortical neurons may adapt in order to dereverberate incoming sounds.

## Reverberation effects result from an adaptive neural process

In principle, there could be other reasons, unrelated to adaptation, why the temporal profile of the inhibitory field is delayed and broader in the more reverberant room. An important possibility is that differences in sound statistics between the reverberation conditions could result in different STRFs, even if the underlying neuronal tuning is unchanged. For example, the cochleagrams of more reverberant sounds are more temporally blurred (**Figure 1C**). This could lead to slower features in the neuronal STRFs for the larger room, purely due to systematic model fitting artefacts (**Christianson**

*et al., 2008*). In combination with changing sound statistics, a non-adaptive static non-linearity in the neural system could produce apparent differences in neuronal tuning between the reverberation conditions (*Christianson et al., 2008*). We therefore performed several additional experiments and analyses to test whether the reverberation-dependent effects observed above are likely to result from a genuine adaptive process.

As a first test, for each recorded cortical unit, we fitted a simulated linear-nonlinear (LN) model neuron (*Schwartz et al., 2006*), composed of a single STRF (fitted to the combined small and large room stimuli) feeding into a non-linear output function (see Simulated neurons). We assessed fit quality using normalized correlation coefficient, $CC_{norm}$ (*Schoppe et al., 2016*), on held-out test data, giving a $CC_{norm}$ value of 0.64. Then a non-homogeneous Poisson process was appended to the LN model, to provide an LNP model. The noise in the recorded neuronal responses was close to Poisson (median Fano factor = 1.1). Since this non-linear model captured the spectrotemporal tuning of the cortical units but did not have an adaptive component, we used it to assess whether our reverberation-dependent results could arise from fitting artefacts in a non-adaptive neuron. To do this, we presented the same stimuli to the simulated non-adaptive neurons as we did to the real neural responses and performed the same analyses. Hence, we fitted STRFs to the simulated neural responses separately for the large and small room conditions. We then extracted *COM* and *PT* parameters from the excitatory and inhibitory temporal profiles of these STRFs, and compared them to those of the measured cortical units. The simulated results are shown alongside the neural results in *Figure 4*.

We asked whether the shift in inhibition observed in the dereverberation model and neural data was also present in this adaptation-free simulation. In the simulation, the inhibitory $COM^-$ was larger for the more reverberant condition (*Figure 4B*), but the effect size for the simulated neurons (median $COM^-$ difference = 4.0ms, p = 9.5 x 10$^{-42}$, Wilcoxon signed-rank test) was less than half of that observed in the real neuronal data (median $COM^-$ difference = 9.3ms, *Figure 4C*). We directly compared the $COM^-$ room differences between cortical units and their simulated counterparts (*Figure 4D*), and found that the reverberation effects on $COM^-$ were consistently larger in the neuronal data (median difference of differences = 5.7ms, mean difference of differences = 6.9ms, p = 7.2 x 10$^{-29}$). An analysis of the peak time of inhibitory STRF components for neural and simulated units was in agreement with the center of mass results (*Figure 4E–G*), with the simulations showing inhibitory peak time shifts (median $PT^-$ difference = 6.4ms, p = 5.9 x 10$^{-32}$) that were more modest than those we observed in the neural data (median difference = 9.4ms, p = 4.0 x 10$^{-44}$). The simulations also showed a modest but significant effect of room size on the excitatory $COM^+$ and $PT^+$ values (*Figure 4B and E*, median $COM^+$ difference = 3.1ms, p = 9.0 x 10$^{-12}$; median $PT^+$ difference = –0.5ms, p = 6.2 x 10$^{-8}$), which was not observed in the neural data (median $PT^+$ difference between neural and simulated data = –2.0ms, p = 4.9 x 10$^{-4}$, median $COM^+$ difference = 0.95ms, p = 6.0 x 10$^{-8}$). When we directly compared the $PT^-$ room differences between cortical units and their simulated counterparts (*Figure 4G*), we found that the reverberation effects on $PT^-$ were consistently larger in the neuronal data (median difference of differences = 1.7ms, mean difference of differences = 10.0ms, p=2.5 x 10$^{-7}$). In summary, these simulations suggest that differences in stimulus properties alone can account for a small shift in inhibitory receptive fields across rooms, but not the magnitude of delay that we observed in our neural data. Therefore, these effects are likely to arise, at least in part, from neural adaptation to room reverberation.

We also investigated the result of replacing the LN component of the LNP model with a model that has a stronger static non-linearity. We used the network-receptive field (NRF) model, which is essentially a single hidden layer neural network, with sigmoid non-linearities for its 10 hidden units and its single output unit (*Harper et al., 2016*; *Rahman et al., 2019*; *Rahman et al., 2020*). We assessed fit quality using $CC_{norm}$ (*Schoppe et al., 2016*) on held-out test data, comparing this to the performance of the LN model. The NRF fits had a mean $CC_{norm}$ of 0.64 and showed statistically significant better performance than the LN fits (median $CC_{norm}$ difference = 0.016, p = 0.0056, Wilcoxon signed-rank test). We repeated the spike rate simulation analyses with this NRF-Poisson (NRFP) model, keeping all other aspects of the analysis the same as described for the LNP model above. As with the LNP model, the NRFP model could not explain the magnitude of the shift in inhibitory center of mass or peak time seen in the real data (*Figure 4—figure supplement 1*). This suggests that an increased non-linearity alone cannot account for the reverberation adaptation observed in auditory cortex.

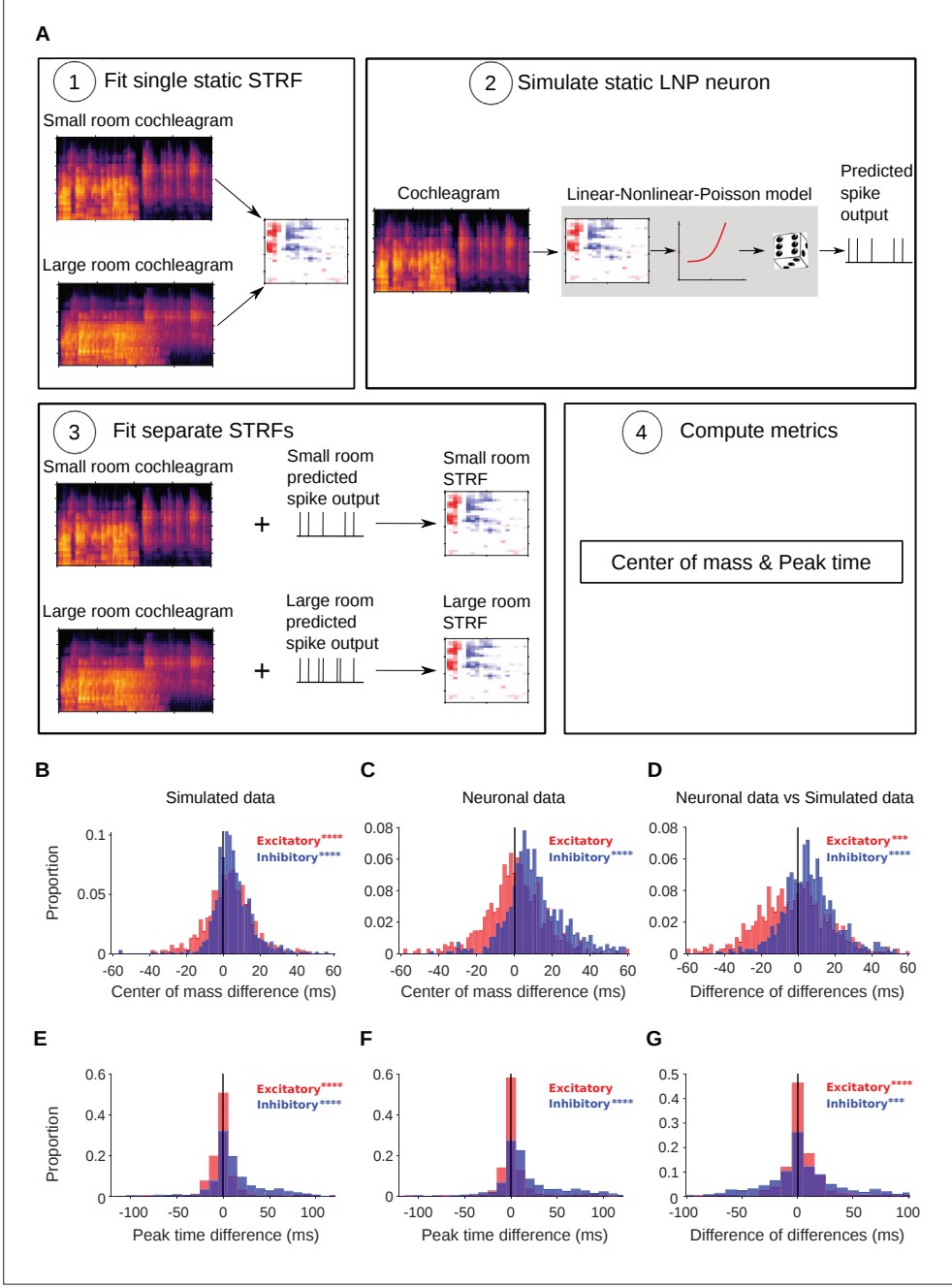

**Figure 4.** Simulated neurons suggest a role for adaptation in cortical dereverberation. To confirm that STRF differences between rooms were genuinely a result of adaptation, we simulated the recorded neurons using a non-adaptive linear-nonlinear-Poisson model and compared STRF measures of the simulated responses with those of the real neuronal STRFs in the different room conditions. (**A**) The simulated neurons were made in the following way: (1) We fitted a single STRF for each neuron using the combined data from the small and large rooms; (2) We used this STRF along with a fitted non-linearity and a Poisson noise model to generate the simulated firing rate for the small and large rooms separately; (3) Using the small and large room cochleagrams and simulated firing rates, we fitted separate STRFs for the two conditions; (4) We computed the center of mass and peak time metrics as before. (**B**) Difference in center of mass between the large and small room conditions (large - small room) for the simulated neurons. The $COM^-$ values (blue) were larger in the large room (median difference = 4.0ms, mean difference = 5.1ms), and the $COM^+$ values (red) were slightly elevated too (median difference = 3.1ms, mean difference = 3.1ms). (**C**) Reproduction of **Figure 3B** showing the difference in center of mass of neuronal STRF components between the large and small room conditions (large - small room). The $COM^-$ values increased in

*Figure 4 continued on next page*

*Figure 4 continued*

the larger room (median difference = 9.3ms, mean difference = 12.0ms), whereas $COM^+$ did not differ significantly (median difference = 0.32ms, mean difference = 0.59ms). (**D**) For each unit, the center of mass differences shown in B were subtracted from those in C and plotted as the resulting difference of $COM$ differences (real cortical unit - simulated neuron). The $COM^-$ differences between rooms were consistently larger in the neuronal data (median difference = 5.7ms, mean difference = 6.9ms), while the $COM^+$ effect was larger in the simulations (median difference = –2.0ms, mean difference = –2.5ms). (**E**) Difference in peak time between the large and small rooms (large - small) for the simulated neurons. The $PT^-$ median difference = 6.4ms (mean difference = 13ms) and the $PT^+$ median difference = –0.50ms (mean difference = –0.43ms). (**F**) Reproduction of *Figure 3D* showing the difference in peak time between the large and small rooms (large - small), calculated from neuronal STRFs. The $PT^-$ values were larger in the large room (median difference = 9.4ms, mean difference = 20.0ms). $PT^+$ did not differ significantly between the rooms (median difference = 0.0ms, mean difference = 3.0ms). (**G**) Histogram of the difference in peak time room differences between the cortical units and corresponding simulated neurons (cortical unit - simulated neuron), plotted as in D above. The $PT^-$ shifts were consistently larger in the neuronal data than in the simulated neurons (median difference = 1.1ms, mean difference = 7.4ms). $PT^+$, on the other hand, showed larger effects of room size in the simulated data (median difference = 0.95ms, mean difference = 3.5ms). Asterisks indicate the significance of Wilcoxon signed-rank tests: ****$p < 0.0001$, ***$p < 0.001$.

The online version of this article includes the following figure supplement(s) for figure 4:

**Figure supplement 1.** Comparison of real neurons and non-adapting network receptive field-Poisson (NRFP) simulated neurons.

To further confirm that the shift in inhibitory receptive fields arises from neuronal adaptation to reverberation and not to differences in stimulus statistics between the room conditions, we compared how all cortical units in our dataset respond to a probe stimulus (a non-reverberated noise burst) interspersed within the small and large room reverberation stimuli (see Noise burst analysis). If the neurons adapt to the current reverberation condition, we should expect them to respond differently to the noise probe when it occurs within the small room and large room stimuli, reflecting the different adaptation states of the neurons. The neuronal responses to the noise probe showed a similar initial onset excitation (5–25ms) in both conditions, but the return to baseline firing was slower in the large room condition (*Figure 5A*). This is consistent with the previous STRF analysis, wherein the excitatory temporal profile was similar between the small and large rooms (*Figure 3B and D*), while the inhibitory components were delayed in time in the large room (*Figure 3B and D*). For each cortical unit, we compared the center of mass of the noise burst response between the small and large rooms (*Figure 5B*). The $COM$ of the noise response increased slightly in the large room (median $COM$ difference = 1.0ms, p = 0.0063). Therefore, responses to an anechoic probe noise show further evidence for reverberation adaptation in auditory cortical neurons, and are consistent with the predicted delayed inhibition in the presence of increased reverberation.

To further confirm and explore the adaptive basis of our results, we presented our reverberant sounds in blocks, which switched between the small and large room every 8s (see *Figure 5C* and Switching stimuli analysis). This switching stimulus was tested in 310 cortical units across 4 ferrets. If the room adaptation accumulates throughout the 8s following a room switch, we would expect the inhibitory component of neuronal STRFs to be decreasingly delayed throughout this period following a switch to the small room and increasingly delayed for a switch to the large room. To test this prediction, we fitted STRFs to neuronal responses separately from the first and last half of each 8s room block, for the small (S1 early and S2 late halves) and large room (L1 early and L2 late halves). The switching stimulus was designed to ensure that the stimulus set of L1 and L2 (or S1 and S2) was the same, but the order of stimuli was shuffled differently for these two time periods. Specifically, we predicted that the neuronal STRFs would have a larger $COM^-$ during the L2 than the L1 period, while $COM^+$ should remain unchanged. By the same reasoning, in a large-to-small room switch, we expected the $COM^-$ to be smaller in S2 than in S1, while $COM^+$ should remain similar.

We observed these predicted trends in our data, as show in *Figure 5D and E*. The $COM^-$ decreased from S1 to S2 (median difference = –0.9ms, Wilcoxon signed-rank test, p = 0.019), while $COM^+$ did not change across these two periods (median difference = 0.52ms, p = 0.85). In the switch to a large room, $COM^-$ increased from the first (L1) to second (L2) half of the block (median difference = 1.5ms, p = 0.0088), while $COM^+$ did not change (median difference = 0.8ms, p = 0.35). These results further

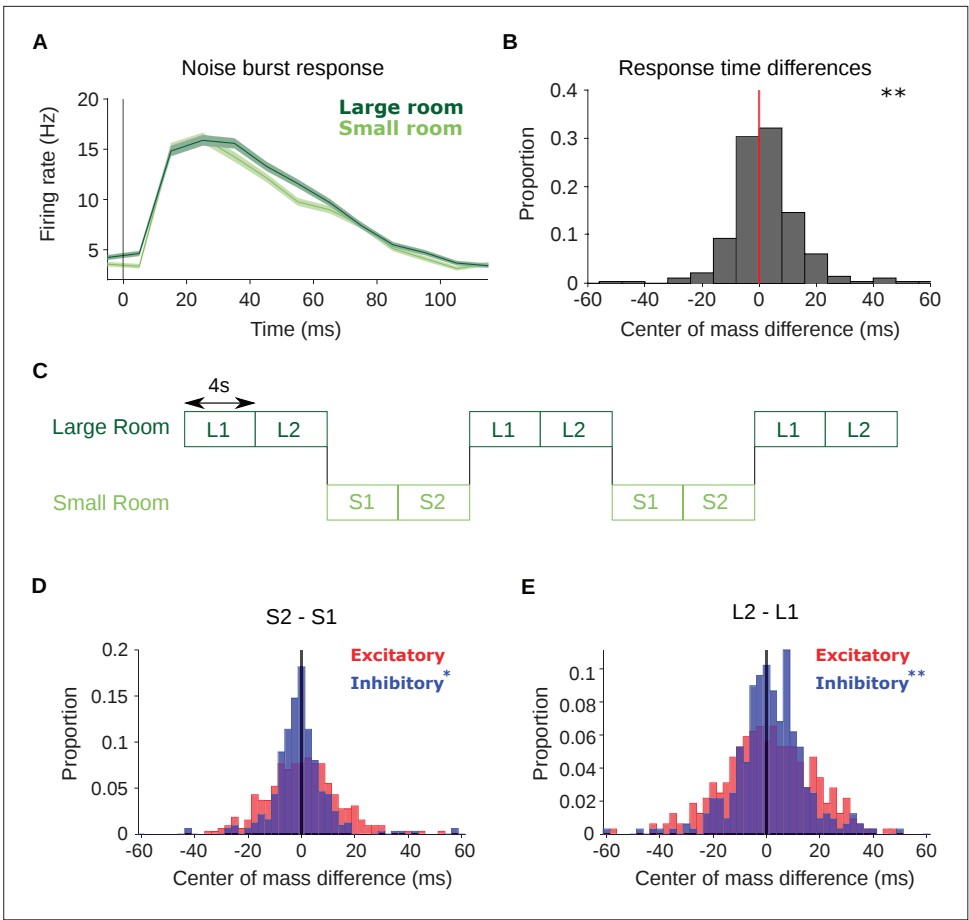

**Figure 5.** Adaptation is confirmed by neural responses to a noise probe and to stimuli that switch between the small and large room. (**A**) Average firing rate across all cortical units in response to an anechoic noise burst that was embedded within the reverberant stimuli. Responses to the noise within the small (light green) and large (dark green) rooms are plotted separately. Shaded areas show ± SEM across units. The vertical line indicates the noise onset. (**B**) Histogram of the difference in center of mass of the neuronal response to the noise probe (shown in A) between the two room conditions (large - small room). The center of mass shifted to a later time in the larger room (median difference = 1.0ms). Asterisks indicate significance of a Wilcoxon signed-rank test: **$p < 0.01$. (**C**) Schematic shows the structure of the 'switching' stimulus, which alternates between the large (dark green) and small room (light green) conditions. Letters indicate the reverberant condition in each stimulus block (S: small room, L: large room). Each 8s block within a given room condition was divided for analysis into an early (S1,L1) and late (S2,L2) period. STRFs were fitted to the data from each of the 4 periods independently (S1, S2, L1, L2). (**D**) Difference in center of mass of inhibitory ($COM^{-}$, blue) and excitatory ($COM^{+}$, red) STRF components between the late and early time period of the small room stimuli (S2 - S1, see A). The $COM^{-}$ decreased in S2 relative to S1 with a median difference = –0.9ms; $COM^{+}$ did not differ significantly, median difference = 0.52ms. (**E**) Center of mass difference plotted as in B, but for the large room stimuli (L2 - L1). The $COM^{-}$ values were larger in L2 relative to L1, median difference = 1.5ms, while the $COM^{+}$ values were not significantly different, median difference = 0.8ms. Asterisks indicate the significance of Wilcoxon signed-rank tests: **$p < 0.01$, *$p < 0.05$.

suggest that auditory cortical receptive fields are genuinely adapting dynamically to the changing reverberant conditions.

## Neural adaptation helps to remove the effects of reverberation

The above results indicate that auditory cortical neurons show adaptation that is consistent with a model of room-dependent dereverberation. To further confirm that the neural adaptation we observed promotes reverberation invariance, we measured the similarity of cortical responses to the same natural sounds across different reverberation conditions. This was compared to the LNP model of the cortical units, which lacks adaptation but approximates each unit's spectrotemporal tuning,

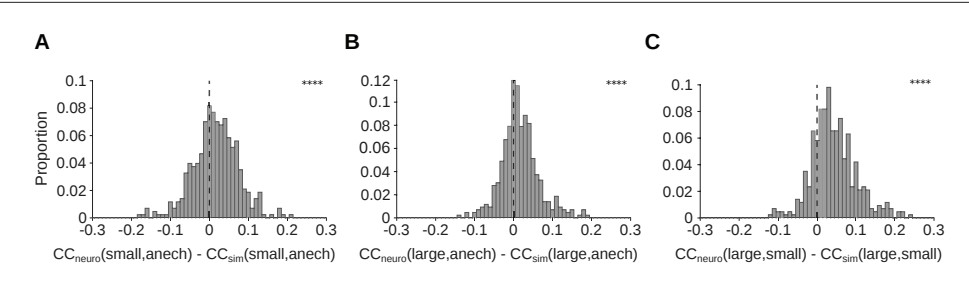

**Figure 6.** Auditory cortical responses are more reverberation invariant than adaptation-free simulated neural responses. Pearson's correlation coefficient (**CC**) was computed between the neural response-over-time (trial-averaged spike count in 10ms time bins) to natural sounds presented in two different reverberant conditions. The correlations for each cortical unit were then compared with the correlation coefficient for the unit's corresponding LNP model. A positive difference between these correlations indicates that the real neuron is more invariant to reverberation than its LNP simulation, suggesting that adaptation may help in removing the effects of reverberation. (**A-C**) Each histogram plots the distribution over units of difference between the correlation coefficient for the recorded neural response-over-time ($CC_{neuro}$) and that for the corresponding simulated response-over-time ($CC_{sim}$; LNP simulations as described in **Figure 4**). (**A**) **CC** difference between recorded and simulated cortical units for the small and anechoic rooms (median difference = 0.016; Z = 6.0; p = 1.5 x 10⁻⁹). (**B**) **CC** difference for the large and anechoic rooms (median difference = 0.012; Z = 6.9; p = 7.2 x 10⁻²). (**C**) **CC** difference for the large and small rooms (median difference = 0.036; Z = 13.0; p = 1.0 x 10⁻⁴⁰). Asterisks indicate the significance of Wilcoxon signed-rank tests: ****$p < 0.0001$.

The online version of this article includes the following figure supplement(s) for figure 6:

**Figure supplement 1.** The estimated cochleagrams produced by the dereverberation model are more reverberation invariant than the original cochleagrams.

output non-linearity, and response variability. We did this for 430 cortical units recorded from 5 ferrets, and included an anechoic room condition. We performed this analysis for three pairs of reverberant conditions: the small room and the anechoic room; the large room and the anechoic room; and the large room and the small room. In all three cases, the real neural responses showed significantly larger correlation coefficients between reverberation conditions than did the simulated neural responses (Wilcoxon signed-rank tests; p<0.0001; **Figure 6**). A similar correlation analysis was used to demonstrate cochleagram dereverberation by our normative model (**Figure 6—figure supplement 1**). These results suggest that the adaptation we observed plays a role in dereverberation by producing neural representations of sounds that are similar across reverberant conditions.

## Frequency dependence of the temporal profile of adaptation

Reverberation is a frequency-dependent effect, as higher frequencies are usually attenuated by air and surfaces faster than lower ones in natural conditions (*Traer and McDermott, 2016*; *Kuttruff, 2017*). Therefore, we explored whether our dereverberation model and auditory cortical neurons also show frequency-dependent reverberation effects.

*Figure 2—figure supplement 1* and *Figure 2—figure supplement 2* plot the reverberation model kernels and neuronal STRFs as a function of their frequency tuning. A visual inspection of these plots reveals that in both the model and the neuronal data, while the temporal spread of the excitatory components stays relatively constant across the preferred frequency, the inhibitory components tend to extend less far back in time as the preferred frequency increases. This temporal narrowing of the inhibitory fields is observed for both the large and the small reverberant rooms. Therefore, the frequency-dependent effects predicted by our dereverberation model are confirmed in our cortical recordings.

To further examine these frequency-tuning effects, we plotted the excitatory and inhibitory center of mass values ($COM^+$, $COM^-$) as a function of the anechoic frequency estimated by the model kernels (**Figure 7A**) or the best frequency of the neuronal STRFs, i.e. the sound frequency of the highest weight (**Figure 7B**). The inhibitory components occurred systematically later in model kernels that were tuned to lower frequencies, in both the small (Pearson's correlation: $r = -0.57$, $p = 0.0037$) and large room ($r = -0.80$, $p = 2.6 \times 10^{-6}$) simulations. The same correlation between best frequency and

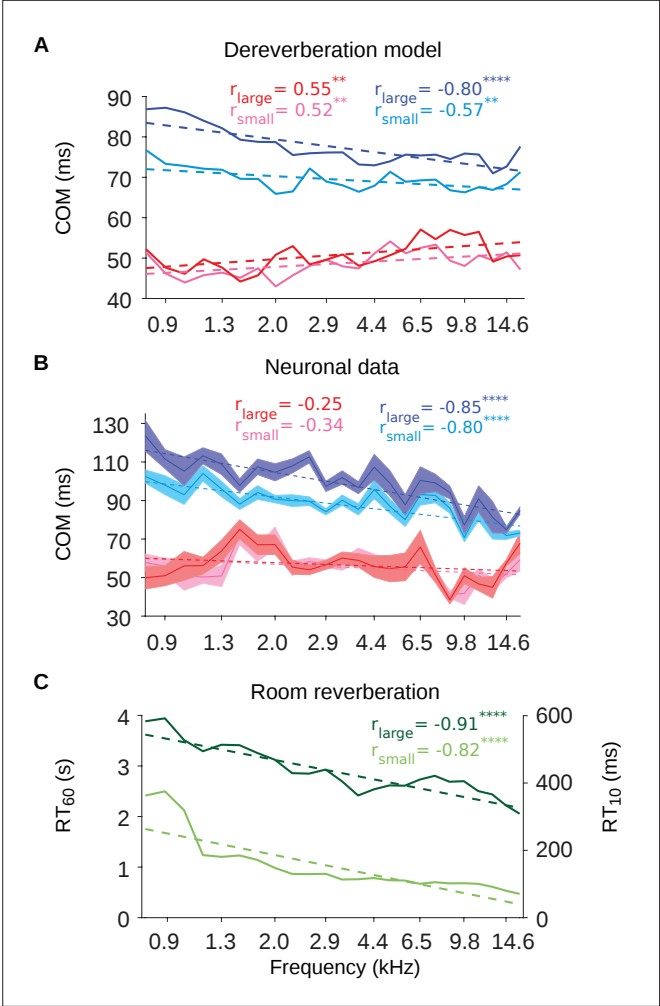

**Figure 7.** The inhibitory tuning latencies and reverberation times show similar frequency dependence. (**A**) Center of mass values (*COM*) are plotted against the anechoic frequency channel being estimated, for the excitatory and inhibitory fields of each model kernel for the large room and for the small room. These are color coded as follows: excitatory *COM* (large room, $COM^+_{large}$, red; small room, $COM^+_{small}$, pink) and their inhibitory counterparts ($COM^-_{large}$, blue; $COM^-_{small}$, cyan). The dashed lines show a linear regression fit for each room, and the Pearson's r value for each fit is given at the top of each the plot. (**B**) *COM* values are plotted against the best frequency for the neuronal data (sound frequency of highest STRF weight). Each cortical unit was assigned a best frequency and the *COM* values measured. The solid lines represent the mean *COM* value for each best frequency, the shaded areas show ± SEM; color scheme and other aspects as in A. (**C**) $RT_{60}$ and $RT_{10}$ values are plotted as a function of cochlear frequency bands, for the large (dark green) and small (light green) rooms. Linear regression fit (dotted line) was used as in A and B to calculate r. Significance of Pearson's correlation: ****$p < 0.0001$, **$p < 0.01$.

The online version of this article includes the following figure supplement(s) for figure 7:

**Figure supplement 1.** Binaural room impulse responses.

$COM^-$ was present in the neuronal STRFs (small room: $r = -0.80$, $p = 3.0 \times 10^{-6}$; large room: $r = -0.85$, $p = 1.6 \times 10^{-7}$). In contrast, the dereverberation model showed a smaller magnitude but significant increase of the excitatory $COM^+$ with best frequency (small room: $r = 0.52$, $p = 0.0087$; large room: $r = 0.55$, $p = 0.0049$), while there was no relationship between $COM^+$ and best frequency in the neuronal data (small room: $r = -0.34$, $p = 0.1$; large room: $r = -0.25$, $p = 0.24$).

*Figure 7A and B* also show that the inhibitory components were later in time in the large room than in the small room across the entire best frequency range, for both the dereverberation model and neuronal data. The $COM^+$ values, on the other hand, were largely overlapping between the two rooms

across this frequency range. This is in agreement with our observations that the inhibitory components of the receptive fields shift reliably with room size, while the excitatory components do not.

The frequency dependence of the inhibitory shift may reflect a frequency dependence in the reverberation acoustics themselves. The decay rate of the power in the impulse response of a reverberant environment depends on sound frequency, and this dependence can change across different environments. However, many man-made and natural environments show a gradual increase in decay rate above about ~0.5kHz (*Traer and McDermott, 2016*). The decay rate can be measured as the reverberation time $RT_{60}$, which is the time necessary for the sound level to decay by 60dB relative to an initial sound impulse (similarly, $RT_{10}$ is the time necessary for a decay by 10dB). The frequency-dependent $RT_{60}$ and $RT_{10}$ values for our small and large rooms are plotted in *Figure 7C*. The impulse responses of both rooms exhibited a decrease in $RT_{60}$ values as a function of frequency (Pearson's correlation, small room: $r = -0.82$, $p = 1.1 \times 10^{-10}$, large room: $r = -0.91$, $p = 8.0 \times 10^{-10}$). This faster decay for higher frequencies can also be observed in the spectrograms of the impulse responses (*Figure 7—figure supplement 1*). Therefore, the frequency-dependent delay in the inhibitory components of our dereverberation model and cortical STRFs paralleled the $RT_{60}$ frequency profile of the virtual rooms in which the sounds were presented.

## Discussion

In this study, we applied a normative modelling approach to ask the question: If a function of the auditory system is to remove reverberation from natural sounds, how might the filtering properties of neurons adapt to achieve this goal? To answer this question, we used a rich dataset of anechoic speech and natural environmental sounds, adding different amounts of reverberation to them. We then trained a linear dereverberation model to remove this reverberation. We constructed our model in such a way that the selectivity (kernels) of the model units after training can be compared to the filtering properties (STRFs) of real auditory cortex neurons in the ferret (*Figure 1*). We confirmed the validity of our dereverberation model by showing that it recapitulated known properties of auditory cortical neurons, such as frequency tuning and temporally asymmetric STRFs with excitation followed by inhibition (*Figure 2*). Interestingly, our dereverberation model also makes two novel predictions: (1) the inhibitory components of neuronal STRFs should be more delayed in more reverberant conditions (*Figure 3*); and (2) the inhibition should occur earlier for higher sound frequencies (*Figure 2—figure supplements 1 and 2 , Figure 7*).

We verified both of these predictions using electrophysiological recordings from ferret auditory cortex neurons, fitting STRFs to neuronal responses to sounds from the same rich dataset, and

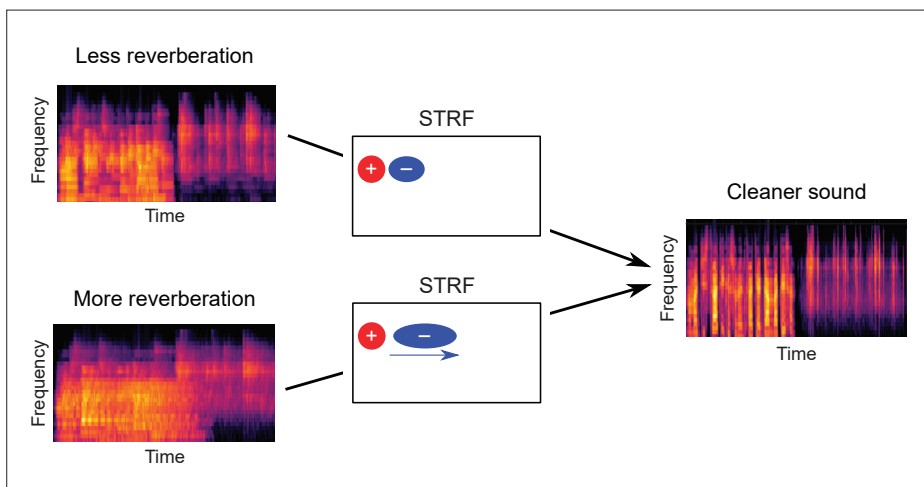

**Figure 8.** Schematic of dereverberation by auditory cortex. Natural environments contain different levels of reverberation (illustrated by the left cochleagrams). Neurons in auditory cortex adjust their inhibitory receptive fields to ameliorate the effects of reverberation, with delayed inhibition for more reverberant environments (center). The consequence of this adaptive process is to arrive at a representation of the sound in which reverberation is reduced (right cochleagram).

comparing them to the model kernels. Finally, we used three additional methods – non-adaptive simulated neurons, probe stimuli and switching stimuli – to confirm that the observed changes in the neuronal STRFs are consistent with a truly adaptive dynamic process (*Figure 4*, *Figure 4—figure supplement 1*, *Figure 5*). Thus, our results suggest that the population of auditory cortex neurons adapt to reverberation by extending their inhibitory field in time in a frequency-dependent manner. This proposed auditory cortical adaptation is summarized in *Figure 8*. In the following, we explore these findings in the broader context of previous studies and possible mechanisms for adaptation to reverberation.

## Auditory cortical neurons adapt their responses to reverberation

Previous studies have shown that human hearing is remarkably robust to reverberation when listeners discriminate speech and naturalistic sounds (*Houtgast and Steeneken, 1985*; *Bradley, 1986*; *Darwin and Hukin, 2000*; *Culling et al., 2003*; *Nielsen and Dau, 2010*). Our neurophysiological results in the ferret auditory cortex are consistent with such robust representation. We find that neurons recorded in the auditory cortex tend to adapt their responses in a way that is consistent with the computational goal of removing reverberation from natural sounds, even in anesthetized animals. Our results are also in good agreement with a previous study in awake passive listening ferrets, which showed that anechoic speech and vocalizations were more readily decodable from the responses of auditory cortex neurons to echoic sounds than the echoic sounds themselves (*Mesgarani et al., 2014*). A similar study in humans using EEG corroborated these findings, showing speech envelopes reconstructed from neural responses to the reverberated stimuli resembled the original anechoic stimuli more than the echoic input, but only when listeners attended to the sound sources (*Fuglsang et al., 2017*).

Interestingly, a human MEG study suggests that auditory cortex may contain both reverberant and dereverberated representations of speech in reverberant conditions (*Puvvada et al., 2017*). In addition, *Traer and McDermott, 2016* found that humans were able to discriminate different reverberant conditions well with both familiar and unfamiliar sounds. In line with this, a minority of neurons in our study did not change the timing of their inhibitory responses in different reverberant conditions or showed the opposite effect from our model prediction (i.e. their $COM^-$ and $PT^-$ decreased in the more reverberant room, *Figure 3B and D*). Thus, although most cortical neurons adapted to reverberation, it is possible that some of them might carry information about the reverberant environment or even represent it more explicitly. The larger variance in reverberation adaptation across neural units may also result from the fact that neural responses are inherently noisier than our model kernels.

## Temporal shifts in inhibition underlie adaptation to reverberation

Our findings build on and provide an explanation for those of *Mesgarani et al., 2014*. These authors approximated a reverberant stimulus by convolving speech and vocalizations with exponentially decaying white noise. In contrast, we used a more diverse stimulus set, which included many environmental sounds that can have very different acoustical statistics (*Attias and Schreiner, 1996*; *Turner, 2010*), and a model of reverberation that included early reflections and their frequency dependence, which are known to have important perceptual effects (*Traer and McDermott, 2016*). *Mesgarani et al., 2014* proposed a combination of subtractive synaptic depression and multiplicative gain change as a potential mechanism for the observed adaptation in their study. However, they acknowledged that other functionally equivalent mechanisms might also be feasible. Notably, their study did not test different echoic conditions with varying amounts of reverberation. Therefore, the time constants of the synaptic depression and gain components in their model were fixed. *Mesgarani et al., 2014* speculated that these time constants might have an important impact in conditions with different amounts of reverberation. This is indeed one of our main novel findings: more reverberant environments require more temporally delayed inhibitory responses within the STRFs of auditory cortical neurons.

## Adaptation to reverberation is frequency dependent

Another novel finding of the present study was that the temporal lag of the inhibition was frequency dependent in both the model kernels and neuronal STRFs (*Figure 2—figure supplements 1 and 2*, *Figure 7*). For both the small and large rooms, the temporal lag of the inhibition, but not the excitation, approximately tracked the reverberant profile over sound frequency of the acoustic spaces (measured by the reverberation times $RT_{60}$ and $RT_{10}$, *Figure 7*). Natural and man-made environments

exhibit certain regularities, and the decline in reverberation over this frequency range is one of them (*Traer and McDermott, 2016*). Future studies could examine whether neurons adapt their responses accordingly to room impulse responses with more unusual reverberation time-frequency profiles.

The frequency dependence of the delay in inhibition likely relates to some degree to the time constants of mean-sound-level adaptation (*Dean et al., 2008*), which also decrease with frequency in inferior colliculus neurons responding to non-reverberant noise stimuli (*Dean et al., 2008*). A study by *Willmore et al., 2016* found that this frequency dependence of mean-sound-level adaptation may impact cortical responses and is consistent with removing a running average from natural sounds. Hence, the frequency dependence we observe in the present study may to some extent reflect general mechanisms for removing both reverberation and the mean sound level, and may be at least partially inherited from subcortical areas.

## Possible biological implementations of the adaptation to reverberation

What might be the basis for the cortical adaptation to reverberation that we have observed? Some plausible mechanisms for altering the inhibitory field include synaptic depression (*David et al., 2009*), intrinsic dynamics of membrane channels (*Abolafia et al., 2011*), hyperpolarizing inputs from inhibitory neurons (*Li et al., 2015*; *Natan et al., 2015*; *Gwak and Kwag, 2020*), or adaptation inherited from subcortical regions such as the inferior colliculus or auditory thalamus (medial geniculate body) (*Dean et al., 2008*; *Devore et al., 2009*; *Willmore et al., 2016*; *Lohse et al., 2020*). Further studies are required to discriminate among these mechanisms, and to determine if the observed reverberation adaptation is subcortical or cortical in origin.

Hence, it would be important to investigate whether the adaptive phenomenon we have found occurs at subcortical levels too, namely the inferior colliculus and the medial geniculate body. Previous research in the inferior colliculus of rabbits has shown that neural responses to amplitude-modulated noise partially compensate for background noise and, for some neurons, particularly when that noise comes from reverberation (*Slama and Delgutte, 2015*). However, this study only examined one room size, so it did not investigate the temporal phenomenon we observed. *Rabinowitz et al., 2013* found that neurons in the inferior colliculus in ferrets generally adapt less to the addition of non-reverberant background noise than those recorded in auditory cortex. This and other studies indicate that an increase in adaptation to sound statistics from auditory nerve to midbrain to cortex helps to construct noise-invariant sound representations in the higher auditory brain (*Dean et al., 2005*; *Dean et al., 2008*; *Watkins and Barbour, 2008*; *Wen et al., 2009*; *Lohse et al., 2020*). However, subcortical adaptation phenomena may be influenced by cortical activity through descending connections (*Robinson et al., 2016*), making it challenging to dissect the neuroanatomical origin of these effects. Similarly, it is possible that reverberation adaptation also becomes more complete as we progress along the auditory pathway.

## Considerations and future work

We undertook our electrophysiological recordings in the present study under general anesthesia in order to control for the effects of attention on reverberation adaptation and to facilitate stable recording of neural responses during our large stimulus set. Cortical adaptation to reverberation has been previously observed in awake listeners (*Mesgarani et al., 2014*; *Fuglsang et al., 2017*), and we observed adaptive inhibitory plasticity in the anesthetized animal that is also consistent with dereverberation. This indicates that this form of adaptation is at least in part driven by stimulus statistics and can occur independently of activity and feedback from higher auditory areas (*Krom et al., 2020*).

Previous work has shown no effect of anesthesia on another kind of adaptation, contrast gain control, in either the ferret auditory cortex (*Rabinowitz et al., 2011*) or the mouse inferior colliculus (*Lohse et al., 2020*). Furthermore, *Khalighinejad et al., 2019* found that adaptation to background noise in human auditory cortical responses was similar whether subjects were actively performing speech-in-noise tasks or were distracted by a visual task. There is therefore no a priori reason to expect that cortical adaptation to reverberation should depend on brain state and be substantially different in awake and anesthetized ferrets. Nevertheless, the effects of attention and behavior on auditory cortical STRFs in the ferret are well documented (*David, 2018*). These can manifest, for example, as gain changes and tuning shifts. Considering the importance of reverberation to perception, it would be interesting to explore the effects described here in behaving animals.

Another point for future research to consider is how our normative model could be further developed. For simplicity and interpretability, we used an elementary linear model. The frequency-dependent suppression observed in our normative model and neuronal receptive fields has relations to the linear frequency-domain approaches to dereverberation used in acoustical engineering (e.g. *Kodrasi et al., 2014*; *Krishnamoorthy and Mahadeva Prasanna, 2009*). However, the performance of such linear dereverberation solutions has limitations, such as when the impulse response changes due to the speaker moving through their environment (*Krishnamoorthy and Mahadeva Prasanna, 2009*). There are more complex and powerful models for dereverberation in acoustical engineering, some of which may provide insight into the biology (*Naylor and Gaubitch, 2010*), and these should be explored in future neurobiological studies. Also, in our modelling we were focused on assessing what characteristics of dereverberation model kernels might change under different conditions, not on how the brain learns to make these changes. Hence, we gave our dereverberation model access to the true anechoic sound, something the brain would not have access to. However, there are blind dereverberation models that aim to dereverberate sounds from just one or two microphones, without access to the original anechoic sounds or room impulse response (*Li et al., 2018*; *Jeub et al., 2010*). These blind dereverberation models will be particularly useful to compare to biology if we want to explore how the brain learns to perform dereverberation with just two ears. It is also worth considering that the auditory system will be performing other functions in addition to dereverberation and these may be useful to add into a model.

## Summary

We have observed in auditory cortical neurons a form of adaptation where the inhibitory component of the receptive fields is delayed as the reverberation time increases in a larger room. This is consistent with the cortex adapting to dereverberate its representation of incoming sounds in a given acoustic space. Dereverberated representations of sound sources would likely be more invariant under different acoustic conditions and thus easier to consistently identify and process, something valuable for any animal's survival. Reverberation is a ubiquitous phenomenon in the natural world and provides a substantial challenge to the hearing impaired and speech recognition technologies. Understanding the adaptive phenomena of the brain that allow us to effortlessly filter out reverberation may help us to overcome these challenges.

## Materials and methods
### Animals
All animal procedures were approved by the local ethical review committee of the University of Oxford and performed under license from the UK Home Office. Three adult female and four adult male ferrets (*Mustela putorius furo*; Marshall BioResources, UK) were used in the electrophysiology experiments (mean age = 8.4 months; standard deviation = 4.2 months).

### Surgical procedure
Terminal electrophysiological recordings were performed on each ferret under general anesthesia. Anesthesia was induced with an intramuscular injection of ketamine (Vetalar; 5mg/kg) and medetomidine (Domitor; 0.02mg/kg), and was maintained with a continuous intravenous infusion of these two drugs in Hartmann's solution with 3.5% glucose and dexamethasone (0.5mg/ml/hr). The animal was intubated and artificially ventilated with medical $O_2$. Respiratory rate, end-tidal $CO_2$, electrocardiogram and blood oxygenation were continuously monitored throughout the recording session. Eye ointment (Maxitrol; Alcon, UK) was applied throughout and body temperature was maintained at 36°C–38°C. Atropine (Atrocare; 0.06mg/kg i.m.) was administered every 6hr, or when bradycardia or arrhythmia was observed.

Once anesthetized, each ferret was placed in a custom-built stereotaxic frame and secured with ear bars and a mouthpiece. After shaving the scalp and injecting bupivacaine (Marcain,<1mg/kg s.c.), the skin was incised and the left temporal muscle removed. A steel holding bar was secured to the skull using dental cement (SuperBond; C&B, UK) and a stainless steel bone screw (Veterinary Instrumentation, UK). A circular craniotomy (10mm diameter) was drilled over the left auditory cortex, and the

dura was removed in this region. The brain surface was covered with a solution of 1.25% agarose in 0.9% NaCl, and silicone oil was applied to the craniotomy regularly throughout recording.

With the ferret secured in the frame, the ear bars were removed, and the ferret and frame were placed in an electrically isolated anechoic chamber for recording. Recordings were then carried out in the left auditory cortex. An Ag/AgCl external reference wire was inserted between the dura and the skull on the edge of craniotomy. A Neuropixels Phase 3 microelectrode probe (*Jun et al., 2017*) was inserted orthogonally to the brain surface through the entire depth of auditory cortex. The cortical area of each penetration was determined based on its anatomical location in the ferret ectosylvian gyrus, the local field potential response latency, and the frequency response area (FRA) shapes of recorded cortical units. Based on these criteria, 95% of the recorded units were either within or on the ventral border of the primary auditory areas (primary auditory cortex, A1, and anterior auditory field, AAF), while the remaining units were located in secondary fields on the posterior ectosylvian gyrus. Following each presentation of the complete stimulus set, the probe was moved to a new location within auditory cortex. Data were acquired at a 30kHz sampling rate using SpikeGLX software (https://github.com/billkarsh/SpikeGLX; *Karsh, 2022*) and custom Matlab scripts (Mathworks).

## Spike sorting

The recorded signal was processed offline by first digitally highpass filtering at 150Hz. Common average referencing was performed to remove noise across electrode channels (*Ludwig et al., 2009*). Spiking activity was then automatically detected and clustered using Kilosort2 software (*Pachitariu et al., 2016*; https://github.com/MouseLand/Kilosort2; *Stringer et al., 2022*). Responses from clusters were manually curated using Phy (https://github.com/cortex-lab/phy; *Bhagat et al., 2022*), and a given cluster was labelled as a single unit if it had a stereotypical spike shape with low variance and its autocorrelation spike histogram showed a clear refractory period. Spikes from a given cluster were often measurable on 4–6 neighboring electrode channels, facilitating the isolation of single units. Only well isolated single units and multi-unit clusters that were responsive to the stimuli (noise ratio <40, *Sahani and Linden, 2003*; *Rabinowitz et al., 2011*) were included in subsequent analyses.

## Sound presentation

Stimuli were presented binaurally via Panasonic RP-HV094E-K earphone drivers, coupled to otoscope speculae inserted into each ear canal. The speculae were sealed in place with Otoform (Dreve Otoplastik). The earphones were driven by a System 3 RP2.1 multiprocessor and headphone amplifier (Tucker-Davis Technologies). Sounds were presented at a sampling rate of 48,828Hz. The output response of the earphones was measured using a Brüel & Kjær calibration system with a GRAS 40DP microphone coupled to the end of the otoscope speculae with a silicone tube. An inverse filter was applied to the speaker output to produce a flat spectral response (±3dB) over the stimulus frequency range (200Hz – 22kHz). Sound intensity was calibrated with an Iso-Tech TES-1356-G sound level calibrator.

## Sound stimuli and virtual acoustic space

There are two stimulus sets, the set used to train the dereverberation model, and the set played to the ferrets, which was prepared from a subset the sounds used to make the first set. The stimuli used to train the dereverberation model were constructed from a dataset consisting of clips of anechoic sounds containing human speech and other natural sounds, such as cracking branches, footsteps, and running water. Most of the sound clips were recorded in an anechoic chamber using a Zoom H2 or Zoom H4 sound recorder, apart from some that came from the RWCP Sound Scene Database in Real Acoustic Environments (*Nakamura et al., 1999*). The clips varied in duration from 3s to 10s. A portion of the clips from the dataset was concatenated together to make a single stimulus of 600s duration. A 0.25s cosine ramp was applied to the onset and offset of each snippet to avoid clipping artifacts in concatenation. The 600s stimulus was then band-pass filtered from 200Hz – 20kHz using an 8th-order Butterworth filter. We also constructed a held-out test set of 100s duration in the same manner using different examples of the same types of sounds from the dataset.

Finally, this stimulus was played in a virtual acoustic space (VAS), providing it with reverberation and head-related filtering. We used the 'Roomsim' software (*Campbell et al., 2005*) to generate the virtual acoustic space. This software creates a cuboidal room of arbitrary x, y and z dimensions and

simulates its acoustic properties for a listener at a particular position and orientation in space, for a sound source at a particular position. The simulations are based on the room-image method (*Allen and Berkley, 1979*; *Heinz, 1993*; *Shinn-Cunningham et al., 2001*). One difference between the standard room-image method and Roomsim is that the latter incorporates the absorption properties of different materials, which can be summarized by their frequency-dependent absorption coefficients. In principle, the amount of reverberation in a room will depend on its size, shape and the material from which the walls are made. For our room simulations, the walls, ceiling, and floor use the frequency-dependent absorption coefficients of stone (*Alvarez-Morales et al., 2014*). We decided to vary the amount of reverberation by changing the room size whilst keeping the other parameters fixed. Four different corridor-shaped rooms were created:

1. Anechoic room
2. Small room (length x width x height, 3m × 0.3m × 0.3m, $RT_{60}$ = 0.78s)
3. Medium room (7.5m × 0.75m × 0.75m, $RT_{60}$ = 1.5s)
4. Large room (15m × 1.5m × 1.5m, $RT_{60}$ = 2.6s)

Thus, processing the 600s stimulus for each room provided four 600s stimuli. Note that the anechoic room does not have a clearly defined 'shape', having no reflecting walls, ceiling or floor, with the acoustic filtering determined only by the relative orientation and distances of the sound source and receiver. Roomsim simulates the orientation-specific acoustic properties of the receiver's head and outer ear, represented by the head-related transfer function (HRTF). In all simulations, we used the same ferret HRTF provided from measurements previously made in the lab on a real ferret (from *Schnupp et al., 2001*). The joint filtering properties of the ferret's HRTF and the room were simulated together by Roomsim to produce a binaural room impulse response (BRIR). The ferret head position and orientation were simulated in the VAS, positioning it 0.15m from the floor, at the midpoint of the room's width (0.15m for the small, 0.375m for the medium and 0.75m for the large) and 1/4 of the room's length from one end (0.75m for the small, 1.875m for the medium and 3.75m for the large) and directly facing the opposite end. In all four room conditions, the sound source was positioned at the same height as the ferret's head (0.15m) and at a distance of 1.5m straight ahead in the direction faced by the ferret (0° azimuth and 0° elevation relative to the ferret's head). The reverberation time $RT_{60}$ is the time necessary for the sound level to decay by 60dB relative to an initial sound impulse, while $RT_{10}$ is the time for the sound level to decay by 10dB. We measured reverberation time using a cochlear model, as explained in the next section Cochlear model.

The stimuli presented to the ferrets were constructed from a representative subset of the anechoic natural stimuli used to train the dereverberation model. We cut 40 different snippets of natural sounds, each 2s in duration, from the clips in the datatset. These 2s snippets were concatenated together into two 40s long stimuli. A 0.25s cosine ramp was applied to the onset and offset of each snippet to avoid clipping artifacts in concatenation. The two 40s stimulus blocks were then processed in VAS in exactly the same way as with the modelling stimulus set, for the same small, medium large and anechoic rooms. This provided two 40s blocks for each reverberant condition (a small, medium, large or anechoic room). We played a stimulus set consisting of the anechoic, small, and large room conditions in five animals and a set consisting of the small, medium and large room conditions in two other animals. The 40s blocks were presented in pseudo random order, with ~5s of silence between blocks. This presentation was repeated ten times, with a different order each time.

## Cochlear model

We used a power-spectrogram ('log-pow') based model of cochlear processing as described in *Rahman et al., 2020*. Briefly, a spectrogram was produced from the sound waveform by taking the power spectrum through a short-time Fourier transform (STFT) using 20ms Hanning windows, with 10ms overlap between adjoining windows. Thus, the time bins were of 10ms duration. The power of adjacent frequency channels was summed using overlapping triangular windows (using code adapted from melbank.m, http://www.ee.ic.ac.uk/hp/staff/dmb/voicebox/voicebox.html) to produce 30 log-spaced frequency channels ranging from 400Hz to 19kHz center frequencies. The resulting power in each channel at each time point was converted to log values and, to avoid log(0), any value below a low threshold (−94dB) was set to that threshold (>99.9% of instances were above this threshold). All analyses involving cochleagrams used the right ear cochleagram, as we recorded from the left auditory cortex.

We used the cochleagram to measure the frequency-band-specific reverberation times ($RT_{60}$ and $RT_{10}$) shown in *Figure 7C*. Our method is similar to that of *Traer and McDermott, 2016*, but for consistency we used our cochlear model rather than theirs. First, we produced an impulse response, the sound induced at the right ear of the ferret in the virtual room by a simple click at the standard source position. Then, we put this impulse response through our cochlear model to generate a cochleagram. Next, for each frequency band in this cochleagram, we fitted a straight line to the plot of the decaying log power output (dB) of the cochleagram over time. Using the slope of this line of best fit, we found the amount of time it took for this output to decay by 60dB for the $RT_{60}$ or by 10dB for the $RT_{10}$. This provided the $RT_{60}$ or $RT_{10}$ for each frequency band. We measured the overall $RT_{60}$ of each room by taking the median $RT_{60}$ over all 30 frequency bands.

## Model kernels

The dereverberation model consisted of a set of linear kernels, one for each of the 30 frequency channels in the anechoic cochleagram. The kernels were fitted separately for each reverberant condition, thus providing 30 kernels for each room. The dereverberation model is summarized by the following equation:

$$\hat{x}^{\text{anech}}_{f't} = \sum_{f=1}^{f_{\max}} \sum_{h=1}^{h_{\max}} k_{f'fh} x^{\text{reverb}}_{f(t-h+1)} + b_{f'} \tag{1}$$

Here, $\hat{x}^{\text{anech}}_{f't}$ is the estimate of the anechoic cochleagram for frequency channel $f'$ and time bin $t$. Obtaining $\hat{x}^{\text{anech}}_{f't}$ involved convolving the kernels $k_{f'fh}$ with the reverberant cochleagram $x^{\text{reverb}}_{ft}$. Here, $f$ is the frequency channel in the reverberant cochleagram and $h$ indexes the time lag used in the convolutions. The model weights $k_{f'fh}$ are composed of 30 kernels, one for each frequency channel $f'$ in the anechoic cochleagram. Finally, the bias term for frequency channel $f'$ is $b_{f'}$.

For each anechoic frequency channel $f'$, the associated model kernel was separately fitted to minimize the mean squared error between the kernel's estimate of that frequency channel of the anechoic cochleagram $\hat{x}^{\text{anech}}_{f't}$ and that actual channel of the anechoic cochleagram $x^{\text{anech}}_{f't}$, subject to $L_2$ regularization ('ridge' regression) on $k_{f'fh}$. The weights were fitted using the glmnet package (GLM, J. Qian, T. Hastie, J. Friedman, R. Tibshirani, and N. Simon, Stanford University, Stanford, CA; http://web.stanford.edu/~hastie/glmnet_matlab/index.html). To select the regularization strength (the hyperparameter $\lambda$), we performed 10-fold cross-validation, using 90% of the data for the training set and 10% (an unbroken 60s segment) for the validation set. Our validation sets over folds were non-overlapping. We found the $\lambda$ that gave the lowest mean-squared error averaged over the 10 folds. Using this $\lambda$, we then re-fitted the model kernels using the whole cross-validation set (training + validation set). These resulting kernels are the ones shown and used in all analyses. These kernels were also used to estimate the dereverberation capacity of the model on the held-out test set. Note that here onward we typically refer to individual model kernels by $k_{fh}$ for brevity, dropping the $f'$ index used for the full set of kernels $k_{f'fh}$.

## Neuronal STRFs

For each cortical unit, for each reverberation condition, we separately estimated its spectro-temporal receptive field (STRF) using its response to the natural stimuli under that condition (*Theunissen et al., 2001*). We used the STRF, a linear model, as this enabled comparison to our linear dereverberation model. The STRF of a cortical unit was the best linear fit from the cochleagram of the stimuli to the unit's response-over-time $y_{nt}$. This response-over-time $y_{nt}$ is the spike counts of cortical unit $n$ over time bins $t$ and was made by counting the spikes in the same 10ms time bins used in the cochleagram and averaging over the 10 stimulus repeats. The STRF model can be summarized by the following equation:

$$\hat{y}_{nt} = \sum_{f=1}^{f_{\max}} \sum_{h=1}^{h_{\max}} w_{nfh} x^{\text{reverb}}_{f(t-h+1)} + b_n \tag{2}$$

Here, $\hat{y}_{nt}$ is the estimated spike counts of cortical unit $n$ at time bin $t$. Also, $x^{\text{reverb}}_{ft}$ is the reverberant cochleagram in frequency channel $f$ and at time $t$. For each unit $n$, the weights in $w_{nfh}$ over frequency channel $f$ and history (time lag) index $h$ provide its STRF. Finally, $b_n$ is the bias term of unit $n$.

Notice the similarity of *Equation 2* to *Equation 1* of the dereverberation model. In both cases, we used the reverberant cochleagram as an input (from either the small, medium, or large room) and fitted the best linear mapping to the output. In the case of neuronal STRFs, the output is the neuronal spike count over time, whereas in the model kernel it is a frequency channel of the anechoic cochleagram. For each cortical unit and room, we separately fitted an STRF by minimizing the mean squared error between the estimated spike counts $\hat{y}_{nt}$ and the observed spike counts $y_{nt}$. To do this, for a given room, we used the first 36s of neural response to the two 40s-stimuli associated with that room (as the last 4s contained a noise probe, see subsection Noise burst analysis). The weights were fitted using the glmnet package (GLM, J. Qian, T. Hastie, J. Friedman, R. Tibshirani, and N. Simon, Stanford University, Stanford, CA; http://web.stanford.edu/~hastie/glmnet_matlab/index.html). As for the model kernels (above), the fitting was subject to $L_2$ regularization. To select the regularization strength (the hyperparameter $\lambda$), we performed 10-fold cross-validation, using 90% of the data for the training set and 10% (an unbroken 7.2s segment) for the validation set. Our validation sets over folds were non-overlapping. We found the $\lambda$ that gave the lowest mean-squared error averaged over the 10 folds. Using this $\lambda$, we then re-fitted the STRFs using the whole cross-validation set (training + validation set). The resulting STRFs are the ones shown and used in all analyses. As with the model kernels, from here onwards we typically refer to an individual STRF for a given cortical unit by the form $w_{fh}$ for brevity, dropping the unit index $n$ used here in $w_{nfh}$.

## Quantification of the temporal effects in model kernels and neuronal STRFs

To quantify the temporal profiles of the model kernels and neuronal STRFs, we chose two different measures:

1. Center of mass ($COM$)
2. Peak time ($PT$)

To compute them, we first obtained the averaged excitatory and inhibitory temporal profiles of the model kernels/neuronal STRFs as follows:

$$v_h^+ = \frac{1}{f_{\max}} \sum_{f=1}^{f_{\max}} [w_{fh}]_+ \tag{3}$$

$$v_h^- = \frac{1}{f_{\max}} \sum_{f=1}^{f_{\max}} [w_{fh}]_- \tag{4}$$

where $w_{fh}$ is the neuronal STRF with $f$ and $h$ subscripts denoting frequency channel and history, respectively. *Equation 3 and 4* are the same for the dereverberation model kernels but with $k$ instead of $w$, as with all subsequent equations in this section. $f_{\max}$ is the number of frequencies (30) in the model kernel/neuronal STRF $w_{fh}$. The notation $[w_{fh}]_+$ and $[w_{fh}]_-$ stand for the element-wise operations $\max(w_{fh},0)$ and $\min(w_{fh},0)$, that is:

$$[\mathrm{w}_{fh}]_+ = \begin{cases} w_{fh} \text{ if } w_{fh} \geq 0 \\ 0 \text{ otherwise} \end{cases} \tag{5}$$

$$[\mathrm{w}_{fh}]_- = \begin{cases} w_{fh} \text{ if } w_{fh} \leq 0 \\ 0 \text{ otherwise} \end{cases} \tag{6}$$

Thus, $v_h^+$ and $v_h^-$ are the frequency-averaged positive-only, $[w_{fh}]_+$, and negative-only, $[w_{fh}]_-$, parts of the kernel/STRF $w_{fh}$.

From this, the $COM$ was defined as follows:

$$COM^+ = \frac{\tau}{\sum_{h=1}^{h_{\max}} v_h^+} \sum_{h=1}^{h_{\max}} (h-0.5)v_h^+ \tag{7}$$

$$COM^- = \frac{\tau}{\sum_{h=1}^{h_{\max}} v_h^-} \sum_{h=1}^{h_{\max}} (h-0.5)v_h^- \tag{8}$$

The duration of a time bin is $\tau = 10$ms, hence time lag in the history of the kernel/STRF ranges from $\tau(h-0.5) = 5$ms to $\tau(h_{\max}-0.5) = 195$ms. Thus, $COM^+$ is the temporal center of mass for the positive

(excitatory) components of the kernel/STRF and $COM^-$ the temporal center of mass for the negative (inhibitory) components.

The peak time ($PT$) was defined as the time at which the excitation and inhibition in the frequency averaged neuronal STRFs/model kernels peaked. Due to the discrete nature of the peak time measure (it can only be a multiple of the time bin size), when measuring it we applied an interpolation (*Akima, 1978*) with a factor of 100 to $v_h^+$ and $v_h^-$ in order to obtain a smoother estimate of peak times. $PT^+$ was taken as the time in $v_h^+$ at which the maximum value occurred, and likewise, $PT^-$ was taken as the time in $v_h^-$ at which the minimum value occurred.

## Simulated neurons

In order to explore whether the changes that we observed are truly adaptive, we used simulated neurons that lacked adaptive receptive fields to generate responses. We then applied the same analyses to these simulated neuronal responses as we did to the actual responses. For each real cortical unit $n$, we constructed a corresponding simulated neuron as a linear-nonlinear-Poisson (LNP) model in the following way. First, we fitted a single STRF as described in section Neuronal STRFs. However, in this case we used the full dataset from the "small" and "large" conditions together, rather than fitting separate STRFs to the two conditions as we did previously.

Next, we fitted a sigmoid output non-linearity by first generating a spike count prediction $\hat{y}_{nt}$ for the full dataset according to *Equation 2* from section Neuronal STRFs, using this single STRF and then finding the sigmoid that best fits (minimizes mean-squared error) the actual spike count $y_{nt}$ according to the following equation:

$$\hat{y}_{nt}^{\mathrm{nonlin}} = \frac{\rho_1}{1+\exp(-(\hat{y}_{nt}-\rho_3)/\rho_2)} + \rho_4 \tag{9}$$

Here, $\hat{y}_{nt}^{\mathrm{nonlin}}$ is the output of the point non-linearity at time bin $t$, providing a new estimate of the cortical unit's spike count. As mentioned, $\hat{y}_{nt}$ is the predicted spike count from the linear stage (see *Equation 2*) at time bin $t$, when fitted to the small and large room responses together. It is the four parameters $\rho_1$, $\rho_2$, $\rho_3$, and $\rho_4$ that are optimized in the fit.

We then used the fitted simulated model to produce an approximation of the real neuronal response-over-time to the reverberant stimulus sets for both the small and large conditions. In order to simulate realistic neuronal noise, we used the $\hat{y}_{nt}^{\mathrm{nonlin}}$ output, at each time bin $t$, as the mean of a Poisson distribution from which we generated 10 'virtual' trials, over which we then averaged to provide the simulated response-over-time. Finally, we performed the same analyses on these simulated neural responses as we did for the real data; we fitted STRFs for the two reverberation conditions separately using these simulated responses in place of the actual responses and then analyzed the resulting STRFs as outlined in the section above (Quantification of the temporal effects in model kernels and neuronal STRFs).

Additionally, we repeated the analysis with the simulated neurons, but replacing the LNP model for each simulated neuron with a network receptive field (NRF) model (*Harper et al., 2016*; *Rahman et al., 2019*; *Rahman et al., 2020*) with an appended inhomogeneous Poisson process, to produce an NRFP model. Specifically, we used the NRF model from *Rahman et al., 2020*, which is essentially a single hidden layer neural network with a sigmoid nonlinearity on the hidden units, the only difference from that model being that we used 10 hidden units. We fitted the NRF model to the neural data by minimizing mean squared error, subject to L1 regularization on the weights. We set the regularization strength using 10-fold cross-validation, as we did for the STRFs in the LN model. For each neuron, after fitting the NRF, it was used as input to an inhomogeneous Poisson process. This was used to simulate neural activity, which was then analyzed in exactly the same way as the LNP model simulated activity.

## Noise burst analysis

To further confirm the adaptive change in properties of neurons across the two reverberant conditions, we presented a 500ms long unreverberated broadband noise burst embedded at a random time in the last 4s of each 40s sound block (i.e. from 36 to 40s) for each condition (small and large). Seven out of the 10 repeats of any stimulus block contained a noise burst, with those seven randomly shuffled within the ten. The random timing distribution of the noise bursts was uniform

and independent across repeats and conditions. For each cortical unit, its response to the noise burst was assessed using a peristimulus time histogram (PSTH) with 10ms time bins. For the majority of units, the firing rate had returned to baseline by 100ms, so we decided to use the 0–100ms time window for further analysis (*Figure 6A*). Different neurons had different response profiles, so in order to compare the adaptive properties in the two conditions we chose the center of mass (*COM*) of the firing rate profile within this window as a robust measure. This was defined similarly to the *COM* measure in subsection Quantification of the temporal effects in model kernels and neuronal STRFs (see also *Equations 7 and 8*). The *COM* for the noise bursts in the large and small conditions was calculated for each cortical unit individually and the difference between the two conditions computed (*Figure 6B*).

### Switching stimuli analysis

In order to confirm and explore the adaptive nature of the neuronal responses to reverberant sounds, we presented "switching stimuli" (*Figure 6C*). These stimuli switched back and forth every 8s between the large room and the small room and were created in the following way. First, we took our original reverberant stimuli for both the small room (80s duration) and large room (80s duration) conditions and divided them into consecutive 4s snippets, providing 20 snippets for each condition. We duplicated these two sets and shuffled each one independently, providing a total of four sets of 20 4s-long snippets. We then combined the snippets into eight 40s-long switching stimuli. These switching stimuli comprised 5 epochs of 8s duration each, with 4 'switches' between the small and large epochs. Half of the stimuli started from the large room condition and the other half from the small room condition. Within each 8s epoch, we defined two periods (period 1: 0–4s and period 2: 4–8s). The large-room periods were denoted by L1 (0–4s) and L2 (4–8s), and the small-room periods by S1 (0–4s) and S2 (4–8s) (*Figure 6C*). The snippets from the first small-room set of 20 snippets populated the 20 S1 periods in order, while those from the second small-room set populated the S2 periods in a different order, due to the shuffling. Likewise, snippets from the first large-room set of 20 snippets populated the 20 L1 periods, and those from the second large-room set populated the L2 periods. Thus, the same set of stimuli were included in S1 and S2, and in L1 and L2, with the only differences being their ordering, and between the small and large room stimuli the amount of reverberation. When the 4s periods and 8s epochs were spliced together, they were cross-faded into each other with a 10ms cosine ramp with 5ms overlap, such that the transition from one period to the next was smooth with no detectable clicks between them. We played the eight 40s stimuli in random order to the ferrets; this was repeated 10 times with the order different each time.

The cortical responses recorded with these stimuli were analyzed using the procedure outlined in subsection Neuronal STRFs. For each cortical unit, we fitted four separate STRFs using the neural responses to the S1, S2, L1, and L2 periods. We did not use the first 8s of each of the eight 40s stimuli, since there was no prior sound (silence) and thus they would not be directly comparable to the other 4 epochs. We also did not use the first 500ms of any of the periods, to avoid potential non-reverberation-related responses from the rapid transitions between them. From the resulting four STRFs, we extracted the $COM^+$ and $COM^-$ values for each and compared S1 to S2, and L1 to L2 (*Figure 6D, E*).

### Code availability

We have provided our Matlab scripts for generating our figures on Github: https://github.com/PhantomSpike/DeReverb; *Ivanov, 2021*. Our neural spiking data are available to download from Dryad: https://doi.org/10.5061/dryad.1c59zw3xv.

### Acknowledgements

We are grateful to Dr Quentin Gaucher for assistance with these electrophysiological experiments. We thank Zlatina Dimitrova for her artwork in Figure 1. We are also grateful to Dr Monzilur Rahman for providing code and advice on analysis. This work was supported by a Wellcome Principal Research Fellowship to AJK (WT108369/Z/2015/Z), a BBSRC New Investigator Award (BB/M010929/1) to KMMW, and a Christopher Welch Scholarship (University of Oxford) to AZI.

## Additional information

### Competing interests

Andrew J King: Senior editor, *eLife*. The other authors declare that no competing interests exist.

### Funding

| Funder | Grant reference number | Author |
| --- | --- | --- |
| Wellcome Trust | WT108369/Z/2015/Z | Andrew J King |
| Biotechnology and Biological Sciences Research Council | BB/M010929/1 | Kerry MM Walker |
| University of Oxford | Christopher Welch Scholarship | Aleksandar Z Ivanov |

The funders had no role in study design, data collection and interpretation, or the decision to submit the work for publication. For the purpose of Open Access, the authors have applied a CC BY public copyright license to any Author Accepted Manuscript version arising from this submission.

### Author contributions

Aleksandar Z Ivanov, Conceptualization, Formal analysis, Investigation, Methodology, Software, Visualization, Writing – original draft, Writing – review and editing; Andrew J King, Conceptualization, Funding acquisition, Project administration, Resources, Supervision, Writing – review and editing; Ben DB Willmore, Nicol S Harper, Conceptualization, Formal analysis, Investigation, Methodology, Software, Supervision, Visualization, Writing – review and editing; Kerry MM Walker, Conceptualization, Funding acquisition, Investigation, Methodology, Supervision, Visualization, Writing – review and editing

### Author ORCIDs

Aleksandar Z Ivanov http://orcid.org/0000-0002-3139-8870
Andrew J King http://orcid.org/0000-0001-5180-7179
Ben DB Willmore http://orcid.org/0000-0002-2969-7572
Kerry MM Walker http://orcid.org/0000-0002-1043-5302
Nicol S Harper http://orcid.org/0000-0002-7851-4840

### Ethics

The animal procedures were approved by the University of Oxford Committee on Animal Care and Ethical Review and were carried out under license from the UK Home Office, in accordance with the Animals (Scientific Procedures) Act 1986 and in line with the 3Rs. Project licence PPL 30/3181 and PIL I23DD2122. All surgery was performed under general anesthesia (ketamine/medetomidine) and every effort was made to minimize suffering.

### Decision letter and Author response

Decision letter https://doi.org/10.7554/eLife.75090.sa1
Author response https://doi.org/10.7554/eLife.75090.sa2

---

## Additional files

### Supplementary files

• Supplementary file 1. Supplementary statistics tables, providing further details of all statistical tests described in this article.

• Transparent reporting form

### Data availability

We have provided our Matlab scripts for generating our model and figures on Github: https://github.com/PhantomSpike/DeReverb, (copy archived at swh:1:rev:83237aa910b696cb82f4a08d50318eaca7e213e9).

The following datasets were generated:

| Author(s) | Year | Dataset title | Dataset URL | Database and Identifier |
|---|---|---|---|---|
| Ivanov A, King AJ, Willmore BD, Walker KM, Harper NS | 2022 | Cortical adaptation to sound reverberation | https://github.com/PhantomSpike/DeReverb | GitHub, PhantomSpike/DeReverb |
| Ivanov A, King AJ, Willmore BD, Walker KM, Harper NS | 2022 | Cortical adaptation to sound reverberation | https://doi.org/10.5061/dryad.1c59zw3xv | Dryad Digital Repository, 10.5061/dryad.1c59zw3xv |

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
