## [Editor Report]

This study identifies a mechanism based on context-dependent plasticity of inhibitory receptive fields that likely plays a role in suppression of reverberation signals in hearing. This new mechanism is a very interesting starting point to describe the biological circuit underpinnings of reverberation suppression, a complex signal processing ability of the auditory system.

---

## [Decision Letter]

**Decision letter after peer review:**

Thank you for submitting your article "Cortical adaptation to sound reverberation" for consideration by *eLife*. Your article has been reviewed by 3 peer reviewers, including Brice Bathellier as the Reviewing Editor and Reviewer #1, and the evaluation has been overseen by Barbara Shinn-Cunningham as the Senior Editor. The following individual involved in review of your submission has agreed to reveal their identity: Nima Mesgarani (Reviewer #3).

Essential revisions:

1. The manuscript is mainly focused on the adaptation of inhibition parameters, and there is no assessment of the efficiency of the dereverberation by cortex. The authors should quantify the similarity of cortical responses for the same sounds with and without reverberation. This measure should be compared to the performance of the dereverberation model (see also comment 4. of reviewer #3).

2. The controls of figure 3 are crucial and should be put in a new main figure, not in the supplements. Especially figure 3 suppl. 2&3.

3. The authors show that a linear receptive field model is not biased by reverberation statistics. However, many papers have shown that auditory processing is non-linear. Therefore, the authors should test this again in a non-linear model (e.g. filterbank followed by a quadratic non-linearity as in the Shamma lab) and improve the cochlear model for example by applying the auditory periphery model described in: Bruce, I. C., Erfani, Y., and Zilany, M. S. A. (2018). "A phenomenological model of the synapse between the inner hair cell and auditory nerve: Implications of limited neurotransmitter release sites," Hearing Research 360:40-54. This and other alternatives are readily available as part of the Auditory Modeling Toolbox (https://www.amtoolbox.org).

Also, it is important to note that dereverberation requires highly nonlinear acoustic processing. The dereverberation algorithms that have been used in speech processing typically try to mask the spectrogram as a method to recover the "direct path" and remove the "reflections." While the resulting "high-pass" filter found in the linear filtering attempts to approximate this nonlinear operation, it is not very effective when used in realistic conditions. While the linear modeling framework here allows the authors to perform a straightforward comparison with auditory receptive fields, this limitation should be noted so the readers are aware of the true difficulty of this task and hence, unexplained mechanisms that remain to be found.

4. The authors should show the energy time curves of the BRIRs at different frequencies and derive the expected adaptation mechanisms already from there. This would greatly simplify the overall concept.

5. A big issue is the differences between the stimuli used to find the receptive field in different reverberant scenarios. While the authors do a good job to show that the differences are not merely due to the statistics of the stimuli, particularly by showing a different response to the probe sound, they cannot claim that "all" of the observed changes are due to adaptation. It is likely reflecting a mix of both, some due to the change in the stimulus correlation and some due to adaptation. Currently, this inherent limitation is not acknowledged.

6. Regarding the discussion of the feedforward/feedback nature of the adaptation to changing background statistics, Khalighinejad et al. also showed that the suppression of background noise is the same when the subject is actively performing speech-in-noise perception and when the subject is distracted by a visual task (Figure 5). Perhaps this observation can strengthen the argument regarding the anesthetized/awake conditions and the nature of the adaptation (lines 447-451). Also, it should be made clear in the discussion that the adaptation phenomenon may not be appearing in cortex, but rather subcortically.

*Reviewer #1 (Recommendations for the authors):*

1. It should be better discussed why the cortex has a much larger variability in adaptation time constants, than the dereverberation model. The authors suggest that it is because cortical neurons are performing other computations. But an alternative explanation could be that there is more noise in the data than in the model.

2. Related to 1, the manuscript is mainly focused on the adaptation of inhibition parameters, and there is no assessment of the efficiency of the dereverberation by cortex. The authors should quantify the similarity of cortical responses for the same sounds with and without reverberation. This measure should be compared to the performance of the dereverberation model.

3. The controls of figure 3 are crucial and should be put in a new main figure, not in the supplements. Especially figure 3 suppl. 2&3.

4. The authors show that a linear receptive field model is not biased by reverberation statistics. However, many papers have shown that auditory processing is non-linear. Therefore, the authors should test this again in a non-linear model.

5. It should be made clear in the discussion that this phenomenon may not be appearing in cortex, but rather subcortically.

*Reviewer #2 (Recommendations for the authors):*

I recommend to show the energy time curves of the BRIRs at different frequencies and derive the expected adaptation mechanisms already from there. This would greatly simplify the overall concept.

As a replacement for the cochleagram, I can recommend to apply the auditory periphery model described in: Bruce, I. C., Erfani, Y., and Zilany, M. S. A. (2018). "A phenomenological model of the synapse between the inner hair cell and auditory nerve: Implications of limited neurotransmitter release sites," Hearing Research 360:40-54. This and other alternatives are readily available as part of the Auditory Modeling Toolbox (https://www.amtoolbox.org).

l.610: How was the "low threshold" defined that was applied to limit the log power values?

l.620: This definition of RT10 is inconsistent with the nomenclature used in ISO standards. There, reverberation time is defined as the time it takes the sound energy to decay by 60 dB, denoted as RT60. To estimate this metric, one can also assume a linear decay and measure only the time it takes to decay, for instance, by 20 dB and then multiply by 3. Still, RT20 is then the result of this extrapolation, not the 20-dB-decay time itself. According to that definition, your reverberation times would need to be multiplied by six. Hence, your large room had a reverberation time of about 2.6 s, similar to a small cathedral.

*Reviewer #3 (Recommendations for the authors):*

This is a very interesting study that tests how the auditory system adapts to reverberant acoustic scenes. The paper is written well, and the results are overall compelling. I have a few comments that hopefully can strengthen the claims of the study.

1. It is important to note that dereverberation requires highly nonlinear acoustic processing. The dereverberation algorithms that have been used in speech processing typically try to mask the spectrogram as a method to recover the "direct path" and remove the "reflections". While the resulting "high-pass" filter found in the linear filtering attempts to approximate this nonlinear operation, it is not very effective when used in realistic conditions. While the linear modeling framework here allows the authors to perform a straightforward comparison with auditory receptive fields, this limitation should be noted so the readers are aware of the true difficulty of this task and hence, unexplained mechanisms that remain to be found.

2. A big issue which the authors are well aware of is the differences between the stimuli used to find the receptive field in different reverberant scenarios. While the authors do a good job to show that the differences are not merely due to the statistics of the stimuli, particularly by showing a different response to the probe sound, they cannot claim that "all" of the observed changes are due to adaptation. It is likely reflecting a mix of both, some due to the change in the stimulus correlation and some due to adaptation. Currently, this inherent limitation is not acknowledged.

3. Related to 2, I think the control experiments that were done to show the changes in the response to the probe sound in different reverberant conditions are worthy of inclusion in the main figures. Perhaps a summary of that can be added to Figure 3, as this is a very important result, without which the observed changes are not very compelling.

4. On the same point, while it is an important observation that the responses to the probe stimulus (non-reverberant) are different in different reverberation contexts, a complementary observation that is missing is the similarity of the cortical responses to varying degrees of reverberation. In other words, if the STRFs change as suggested to produce a less variable response to reverberation, then the responses to the same sound with different reverb should stay constant and similar to the anechoic sound. While invariant cortical responses have been shown in other studies (including ferrets), it will strengthen this study if they can also confirm the presence of that effect which is the subject of this study.

5. Regarding the discussion of the feedforward/feedback nature of the adaptation to changing background statistics, Khalighinejad et al. also showed that the suppression of background noise is the same when the subject is actively performing speech-in-noise perception and when the subject is distracted by a visual task (Figure 5). Perhaps this observation can strengthen the argument regarding the anesthetized/awake conditions and the nature of the adaptation (lines 447-451).

---

## [Author Response]

Essential revisions:1. The manuscript is mainly focused on the adaptation of inhibition parameters, and there is no assessment of the efficiency of the dereverberation by cortex. The authors should quantify the similarity of cortical responses for the same sounds with and without reverberation. This measure should be compared to the performance of the dereverberation model (see also comment 4. of reviewer #3).

Our understanding of this request is that the reviewers would like us to examine whether the cortex effectively removes reverberations from its representations of sound, as shown previously in the awake ferret auditory cortex using decoding approaches (Mesgarani et al., 2014). The reviewers suggest that we quantify the similarity of cortical responses to the same sounds across reverberant conditions. To achieve this, we could quantify the similarity of response for each cortical unit by calculating the correlation coefficient between the spiking responses (PSTHs) in the reverberant and anechoic conditions. However, to determine how effective the dereverberation has been, the resulting correlation must be compared to some baseline. The reviewers suggest that we use our normative dereverberation model as a baseline. However, the normative model maps from echoic cochleagrams to anechoic cochleagrams, and so does not produce a spike train with which we could make a direct comparison to neural spiking responses. We therefore need to find an alternative baseline.

Elsewhere in the paper, we have used a non-adapting LNP model as a baseline for comparison with the neural data (the simulated model neurons described in Figure 4). Since this model describes the responses of each cortical neuron in a way that does not include adaptation, it also provides a valid baseline for evaluation of the extent to which cortical adaptation has resulted in dereverberation. If the auditory cortex removes reverberations from its response to sounds, the correlation coefficients between cortical responses to sounds with and without reverberation will be higher than the same metrics calculated with the simulated responses. We would also expect the neuronal responses to be more similar than the model responses to sounds presented in rooms with different levels of reverberation, so we also examine the similarity of spiking responses between the small and large rooms.

The results of these analyses are plotted below. We find that the spiking responses of cortical units across the 3 different reverberant environments are more similar than those of their corresponding LNP simulated model responses. The measured cortical responses were more similar than simulated responses across the small and anechoic rooms (Wilcoxon signed-rank tests; Z = 6.0; p = 1.5 x 10^-9^), the large and anechoic rooms (Z = 6.9; p = 7.2 x 10^-12^), and the small and large rooms (Z = 13.0; p = 1.0 x 10^-40^). These analyses suggest that as a result of adaptive changes in the STRFs, the auditory cortex does effectively remove some of the effects of reverberation from its responses to natural sounds. We have added the figure of these results (below) to our manuscript as Figure 6, and describe it in the results text copied below.

Lines 323-338:

“Neural adaptation helps to remove the effects of reverberation

The above results indicate that auditory cortical neurons show adaptation that is consistent with a model of room-dependent dereverberation. To further confirm that the neural adaptation we observed promotes reverberation invariance, we measured the similarity of cortical responses to the same natural sounds across different reverberation conditions. This was compared to the LNP model of the cortical units, which lacks adaptation but approximates each unit’s spectrotemporal tuning, output nonlinearity, and response variability. We did this for 430 cortical units recorded from 5 ferrets, and included an anechoic room condition. We performed this analysis for three pairs of reverberant conditions: the small room and the anechoic room; the large room and the anechoic room; and the large room and the small room. In all three cases, the real neural responses showed significantly larger correlation coefficients between

reverberation conditions than did the simulated neural responses (Wilcoxon signed-rank tests; p < 0.0001; Figure 6). A similar correlation analysis was used to demonstrate cochleagram dereverberation by our normative model (Wilcoxon signed-rank tests; p < 0.0001; Figure 6 —figure supplement 1). These results suggest that the adaptation we observed plays a role in dereverberation by producing neural representations of sounds that are similar across reverberant conditions.”

We performed a similar analysis on the dereverberation model by measuring the correlation between cochleagram channel activities (rather than correlations between spike responses). As with the neural data, we found that the dereverberation model also leads to increased invariance across reverberant conditions, because the dereverberation process of the model renders the reverberant cochleagram more similar to the anechoic cochleagrams and to each other. We now include this analysis as a supplementary figure (Figure 6 —figure supplement 1).

2. The controls of figure 3 are crucial and should be put in a new main figure, not in the supplements. Especially figure 3 suppl. 2&3.

We have implemented this suggestion to move the control figures into the main manuscript. The previous Supplementary Figure 3-2, which describes the results of our simulated neuron, is now Figure 4. Similarly, the noise burst control (previous Supp Figure 3-3) and room switching time course analysis (previous Supp Figure 3-4) are now combined in Figure 5 of our main manuscript. The references to figure numbers have been updated throughout.

3. The authors show that a linear receptive field model is not biased by reverberation statistics. However, many papers have shown that auditory processing is non-linear. Therefore, the authors should test this again in a non-linear model (e.g. filterbank followed by a quadratic non-linearity as in the Shamma lab)

In our original simulated neurons analysis, we showed that a static linear-nonlinear receptive field model in combination with reverberation statistics cannot explain the size of the temporal shifts in inhibition that we see. We have edited the results to further emphasize the non-linearity of this model, so that it is clearer to the reader.

Lines 241-242:

“Since this nonlinear model captured the spectrotemporal tuning of the cortical units but did not have an adaptive component,…”

However, we agree that it would be interesting to explore if the adaptive inhibitory shift we observed could be explained by a stronger form of static non-linearity, as we mentioned in our Discussion. We could not find a reference for the quadratic non-linearity model of the Shamma lab that the Reviewer mentioned, so we have opted instead to use the Network Receptive Field (NRF) model. This model is substantially more nonlinear than the LN model and consists of a single hidden layer neural network. It has been shown to improve predictions of auditory cortical responses over an LN model (Harper et al. 2016).

We replaced the LN model in our simulated neurons analysis with an NRF model, and otherwise repeated the simulation analysis in exactly the same way. As with the LN model, we found that the NRF model, which had nonlinearity both before and after the final integration, could not explain the size of the temporal shifts in inhibition that we see. This further argues that we see a genuine adaptive effect. We have added these NRF simulations to the manuscript in the following text.

Lines 273-284:

“We also investigated the result of replacing the LN component of the LNP model with a model that has a stronger static non-linearity. We used the network-receptive field (NRF) model, which is essentially a single hidden layer neural network, with sigmoid nonlinearities for its 10 hidden units and its single output unit (Harper et al. 2016; Rahman et al. 2019; 2020). We assessed fit quality using CC_norm_ (Schoppe et al. 2016) on held-out test data, comparing this to the performance of the LN model. The NRF fits had a mean CC_norm_ of 0.64 and showed statistically significant better performance than the LN fits (median CC_norm_ difference = 0.016, p = 0.0056, Wilcoxon signed-rank test). We repeated the spike rate simulation analyses with this NRFPoisson (NRFP) model, keeping all other aspects of the analysis the same as described for the LNP model above. As with the LNP model, the NRFP model could not explain the magnitude of the shift in inhibitory center of mass or peak time seen in the real data (Figure 4—figure supplement 1). This suggests that an increased non-linearity alone cannot account for the reverberation adaptation observed in auditory cortex.”

Lines 784-794:

“Additionally, we repeated the analysis with the simulated neurons, but replacing the LNP model for the simulated neuron with a network receptive field (NRF) model (Harper et al. 2016; Rahman et al. 2019; 2020) with an appended inhomogeneous Poisson process, to produce an NRFP model. Specifically, we used the NRF model from Rahman et al. (2020), which is essentially a single hidden layer neural network with a sigmoid nonlinearity on the hidden units, the only difference being that we used 10 hidden units. We fitted the NRF model to the neural data by minimizing mean squared error, subject to L1 regularization on the weights. We set the regularization strength using 10-fold cross-validation, as we did for the STRFs in the LN model. For each neuron, after fitting the NRF, it was used as input to an inhomogeneous Poisson process. This was used to simulate neural activity, which was then analyzed in exactly the same way as the LNP model simulated activity.”

Relating to this point, we also added an estimate of the Fano factor of the neural responses to demonstrate that the neural noise was approximately Poisson, validating our assumption of Poisson noise in both our model simulations. We have also included a metric to quantify the quality of fit (CC_norm_) of the LN and NRF models, so that the reader can compare these two models in terms of their ability to predict our neural data:

Lines 237-243:

“We assessed fit quality using normalized correlation coefficient, CC_norm_ (Schoppe et al., 2016), on held-out test data, giving a CC_norm_ value of 0.64. Then a non-homogeneous Poisson process was appended to the LN model, to provide an LNP model. The noise in the recorded neuronal responses was close to Poisson (median Fano factor = 1.1). Since this nonlinear model did not have an adaptive component, we used it to assess whether our reverberation-dependent results could arise from fitting artefacts in a non-adaptive neuron.”

Related revision: investigating our LN model simulations for these analyses helped us to uncover a bug in our original code, in which the small room was used to simulate both responses in step 3 of Figure 4. We have now fixed the bug and re-run these simulations in the present version. Following this correction, the inhibitory shifts in our neural responses remained substantially larger than those observed with the simulated responses, as originally reported.

Related revision: Please note that when measuring peak time (PT) that we also now interpolate our frequency averaged neuronal STRFs/model kernels to provide a more resolved estimate of the peak time (PT). This is explained in the methods, copied below. Median PT values are updated throughout the manuscript, but these do not change our overall pattern of results or interpretations of them.

Lines 755-759: “Due to the discrete nature of the peak time measure (it can only be a multiple of the time bin size), when measuring it we applied an interpolation (Akima, 1978) with a factor of 100 to *v_H_^+^* and *v_h_^-^* in order to obtain a smoother estimate of peak times. PT^+^ was taken as the time in *v_H_^+^* at which the maximum value occurred, and likewise, PT^-^ was taken as the time in *v_h_^-^* at which the minimum value occurred.”

The author should improve the cochlear model for example by applying the auditory periphery model described in: Bruce, I. C., Erfani, Y., and Zilany, M. S. A. (2018). "A phenomenological model of the synapse between the inner hair cell and auditory nerve: Implications of limited neurotransmitter release sites," Hearing Research 360:40-54. This and other alternatives are readily available as part of the Auditory Modeling Toolbox (https://www.amtoolbox.org).

We chose our cochlear model after careful consideration. In a recent study (Rahman et al., 2020), we examined which cochlear model provides the best preprocessing stage for predicting neural responses to natural sounds in the primary auditory cortex of ferrets on a held out dataset when the cochlear model is used as input to a linear or non-linear receptive field model (e.g. an LN model). We tested a wide range of cochlear models, including the one that we have used here (referred to as the “*log-pow”* cochlear model in Rahman et al. 2020) and the model of Bruce et al. (2018). The model which performed best was the log-pow model; this was the case across different stimulus types, for awake and anesthetized ferrets, and for linear and non-linear cortical receptive field models. This is why we chose to use the log-pow cochlear model here.

The Bruce et al. (2018) cochlear model, while useful for modeling cochlear responses themselves, did not perform as well as the log-pow model as a preprocessing stage for modeling auditory cortical responses to natural sounds. In Rahman et al. (2020), we speculate that this may be due to brainstem filtering removing many of the details of cochlear processing, rendering the simpler log-pow model more predictive of cortical responses. Thus, in the present paper we have kept the log-pow model as our primary cochlear model, and we hope the Editor and Reviewers will agree that this is the best choice given the results of Rahman et al. (2020). We now explain our choice of cochlear model in the Results:

Lines 90-94: “We used a simple “log-pow” cochlear model to produce the cochleagrams, as our recent work suggests that these cochleagrams enable better prediction of cortical responses than more detailed cochlear models (*Rahman et al., 2020*). These spectrotemporal representations of the sound estimate the cochlear filtering and resulting representation of the sound at the auditory nerve (Brown and Cook, 1994; Rahman et al., 2020).”

However, we agree that it is valuable to use more than one cochlear model to check the robustness of our results. To this end, we have re-calculated the main results of our study using the Bruce et al. (2018) cochlear model (also developed by Carney and colleagues in Zilany et al., 2014, 2009), as per the Reviewer’s suggestion. We refer to this model as the Carney Bruce Erfani Zilany (CBEZ) model. Our central findings regarding the inhibitory time shifts were replicated with this alternative cochlear model, and we have incorporated these analyses into the paper as a supplementary figure (Figure 3 —figure supplement 1, see below). Using the CBEZ model, we observed a shift in the center of mass of inhibitory components (COM^-^) to later time points in the larger room, in line with our original findings, for both the dereverberation model (median difference of 10.0ms for large – small room; Wilcoxon signed-rank test; p = 1.7 x 10^-6^) and neural units (median = 12.0ms; p = 4.2 x 10^-78^). Similarly, the peak time (PT^-^) of inhibitory components also showed a delay in the larger room, for both the dereverberation model (median = 21.0ms; p = 4.7 x 10^-6^) and neural units (median = 12ms; p = 4.6 x 10^-58^). It is worth noting that with the CBEZ model we observed relatively small shifts in the excitatory component COM (COM^+^) towards earlier values in the larger room for the dereverberation model (median difference = -5.6ms; large – small room; p = 1.6 x 10^-5^), and towards later COM^+^ values in the larger room for the neural responses (median = 2.7ms; p = 1.2 x 10^-7^). However, the peak times of excitatory responses showed no difference between rooms in the dereverberation model (PT^+^ median = 0.0ms; p = 1), and a small but statistically significant shift towards later peak times in the larger room in the neural data (PT^+^ median = 0.3ms; p = 6.2 x 10^-9^). Therefore, excitatory shifts were small and inconsistent in both the log-pow and CBEZ models.

Overall, both cochlear models showed a robust delay in the negative components of model kernels and neuronal STRFs in the larger room, with modest or absent effects of room size on positive components. Given the results of Rahman et al. (2020), we have continued to use the simpler log-pow cochlear model in our main manuscript. We have included the CBEZ analysis as a supplementary figure, and refer to it in our main results text:

Lines 187-190: “We also observed these room-size-dependent delays in the COM and PT of inhibitory components when we used a more detailed cochlear model (*Bruce et al., 2018; Zilany et al., 2014; 2009*) to generate input cochleagrams (Figure 3—figure supplement 1).”

Also, it is important to note that dereverberation requires highly nonlinear acoustic processing. The dereverberation algorithms that have been used in speech processing typically try to mask the spectrogram as a method to recover the "direct path" and remove the "reflections." While the resulting "high-pass" filter found in the linear filtering attempts to approximate this nonlinear operation, it is not very effective when used in realistic conditions. While the linear modeling framework here allows the authors to perform a straightforward comparison with auditory receptive fields, this limitation should be noted so the readers are aware of the true difficulty of this task and hence, unexplained mechanisms that remain to be found.

We agree that it is important to make the limitations of linear dereverberation clear to the reader. We have expanded the text that addresses this issue in the Discussion. Our new text is highlighted in yellow below.

Lines 509-518: “Another point for future research to consider is how our normative model could be further developed. For simplicity and interpretability, we used an elementary linear model. The frequency-dependent suppression observed in our normative model and neuronal receptive fields has relations to the linear frequency-domain approaches to dereverberation used in acoustical engineering (e.g., Kodrasi et al. (2014), Krishnamoorthy and Mahadeva Prasanna (2009)). However, the performance of such linear dereverberation solutions has limitations,

such as when the impulse response changes due to the speaker moving through their environment (Krishnamoorthy and Mahadeva Prasanna, 2009). There are more complex and powerful models for dereverberation in acoustical engineering, some of which may provide insight into the biology (Naylor and Gaubtich, 2010), and these should be explored in future neurobiological studies.”

We also now emphasize that our dereverberation model is a simple one, in the first line of our Results.

Lines 79-80: “We trained a simple dereverberation model to estimate the spectrotemporal structure of anechoic sounds from reverberant versions of those sounds.”

4. The authors should show the energy time curves of the BRIRs at different frequencies and derive the expected adaptation mechanisms already from there. This would greatly simplify the overall concept.

We agree it would be valuable to show the energy time curves of the BRIRs, and we now do this in Figure 7—figure supplement 1 (below). We reference this figure in the manuscript with the following text:

Lines 379-380: “This faster decay for higher frequencies can also be observed in the spectrograms of the impulse responses (Figures 7—figure supplement 1).”

As our analyses were based on monaural cochleagrams, we now specify that the right ear input was used:

Lines 667-668: “All analyses involving cochleagrams used the right ear cochleagram, as we recorded from the left auditory cortex.”

BRIR-based dereverberation has made important contributions to acoustical engineering. However, the goal of our manuscript is not to develop or test methods to achieve the best room dereverberation. Rather, our goal is to understand how neurons in the auditory cortex may achieve partial dereverberation through adaptive processes.

To this end, we built our normative model to have structures that resemble simplified versions of processing in the auditory system, in order to force brain-like constraints on the model and to enable comparison to neural data. This includes increased frequency bandwidth at higher frequencies, loss of phase information, substantial compression of intensity in the cochlear model, and an STRF-like linear process in the dereverberation model. Our simple model is far from a complete account of dereverberation in the auditory system, but it is sufficient to provide insights into the questions that we ask.

It is unclear to us how results that are easily comparable to neural data could be derived directly from the BRIR energy-time curves, and it seems to us easiest to instead fit a neural-processinglike model to the dereverberation task at hand as we have done, and then compare that to the neural data. The form of our normative model provides us with STRF-like kernels that can be compared directly with neural STRFs.

5. A big issue is the differences between the stimuli used to find the receptive field in different reverberant scenarios. While the authors do a good job to show that the differences are not merely due to the statistics of the stimuli, particularly by showing a different response to the probe sound, they cannot claim that "all" of the observed changes are due to adaptation. It is likely reflecting a mix of both, some due to the change in the stimulus correlation and some due to adaptation. Currently, this inherent limitation is not acknowledged.

We agree. This interpretation is supported by the results of our simulated neurons, which show some, albeit smaller, systematic shifts in their receptive fields in response to sounds presented in the two different rooms. In the case of our noise burst control, the effects are in fact all due to adaptation, as the noise burst stimulus is identical in both reverberation contexts (small and large room). Similarly for our room switching control, the effects must be due to adaptation, as the stimuli are identical in L1 and L2 and in S1 and S2. Therefore, as the Reviewers point out, while most of the effects in our main analysis are likely to reflect neural adaptation, stimulus statistics are likely to contribute as well.

We have reworded our interpretation of our simulation results to make this point clearer.

Lines 269-272: “In summary, these simulations suggest that differences in stimulus properties alone can account for a small shift in inhibitory receptive fields across rooms, but not the magnitude of delay that we observed in our neural data. Therefore, these effects are likely to arise, at least in part, from neural adaptation to room reverberation.”

6. Regarding the discussion of the feedforward/feedback nature of the adaptation to changing background statistics, Khalighinejad et al. also showed that the suppression of background noise is the same when the subject is actively performing speech-in-noise perception and when the subject is distracted by a visual task (Figure 5). Perhaps this observation can strengthen the argument regarding the anesthetized/awake conditions and the nature of the adaptation (lines 447-451).

We thank the Reviewers for bringing this relevant paper to our attention, and have amended our discussion to include it.

Lines 499-505: “Previous work has shown no effect of anesthesia on another kind of adaptation, contrast gain control, in either the ferret auditory cortex (*Rabinowitz et al., 2011*) or the mouse inferior colliculus (*Lohse et al., 2020*). Furthermore, *Khalighinejad et al. (2019)* found that adaptation to background noise in human auditory cortical responses was similar whether subjects were actively performing speech-in-noise tasks or were distracted by a visual task. There is therefore no a priori reason to expect that cortical adaptation to reverberation should depend on brain state and be substantially different in awake and anesthetized ferrets.”

Also, it should be made clear in the discussion that the adaptation phenomenon may not be appearing in cortex, but rather subcortically.

The Reviewers are correct in pointing out that the effects we have observed in auditory cortex could be subcortical in origin. We make this point in two parts of our Discussion.

First, in the context of our frequency dependence effect, where we have now strengthened the wording:

Lines 463-465: “Hence, the frequency dependence we observe in the present study may to some extent reflect general mechanisms for removing both reverberation and the mean sound level, and may be at least partially inherited from subcortical areas.”

Second, we raise this point in our discussion of potential mechanisms. We have also edited this text to make the point about potential subcortical contributions clearer:

Lines 467-474: “What might be the basis for the cortical adaptation to reverberation that we have observed? Some plausible mechanisms for altering the inhibitory field include synaptic depression (David et al., 2009), intrinsic dynamics of membrane channels (Abolafiaetal., 2011), hyperpolarizing inputs from inhibitory neurons (Li et al., 2015; Natan et al., 2015; Gwak and Kwag, 2020), or adaptation inherited from subcortical regions such as the inferior colliculus or auditory thalamus (medial geniculate body) (Dean et al., 2008; Devore et al., 2009; Willmore et al., 2016; Lohse et al., 2020). Further studies are required to discriminate among these mechanisms, and to determine if the observed reverberation adaptation is subcortical or cortical in origin.

Reviewer #1 (Recommendations for the authors):1. It should be better discussed why the cortex has a much larger variability in adaptation time constants, than the dereverberation model. The authors suggest that it is because cortical neurons are performing other computations. But an alternative explanation could be that there is more noise in the data than in the model.

We have now acknowledged this point in the text.

Lines 429-431: “The larger variance in reverberation adaptation across neural units may also result from the fact that neural responses are inherently noisier than our model kernels.”

2. Related to 1, the manuscript is mainly focused on the adaptation of inhibition parameters, and there is no assessment of the efficiency of the dereverberation by cortex. The authors should quantify the similarity of cortical responses for the same sounds with and without reverberation. This measure should be compared to the performance of the dereverberation model.

See response to Essential revision 1 above.

3. The controls of figure 3 are crucial and should be put in a new main figure, not in the supplements. Especially figure 3 suppl. 2&3.

See response to Essential Revision 2 above.

4. The authors show that a linear receptive field model is not biased by reverberation statistics. However, many papers have shown that auditory processing is non-linear. Therefore, the authors should test this again in a non-linear model.

See response to Essential revision 3 above.

5. It should be made clear in the discussion that this phenomenon may not be appearing in cortex, but rather subcortically.

See response to Essential revision 6 above.

Reviewer #2 (Recommendations for the authors):I recommend to show the energy time curves of the BRIRs at different frequencies and derive the expected adaptation mechanisms already from there. This would greatly simplify the overall concept.

See response to Essential revision 4 above.

As a replacement for the cochleagram, I can recommend to apply the auditory periphery model described in: Bruce, I. C., Erfani, Y., and Zilany, M. S. A. (2018). "A phenomenological model of the synapse between the inner hair cell and auditory nerve: Implications of limited neurotransmitter release sites," Hearing Research 360:40-54. This and other alternatives are readily available as part of the Auditory Modeling Toolbox (https://www.amtoolbox.org).

See response to Essential revision 3 above.

l.610: How was the "low threshold" defined that was applied to limit the log power values?

The threshold was used to avoid the problem that log(0) = -infinity. The threshold was set to 94dB. This value ensures that almost all the output of the log function is above threshold. We have now added this threshold value to our methods section. In practice, over 99.9% of our cochleagram values were above the threshold of -94dB.

Lines 664-667: “The resulting power in each channel at each time point was converted to log values and, to avoid log(0), any value below a low threshold (-94dB) was set to that threshold (>99.9% of instances were above this threshold).”

l.620: This definition of RT10 is inconsistent with the nomenclature used in ISO standards. There, reverberation time is defined as the time it takes the sound energy to decay by 60 dB, denoted as RT60. To estimate this metric, one can also assume a linear decay and measure only the time it takes to decay, for instance, by 20 dB and then multiply by 3. Still, RT20 is then the result of this extrapolation, not the 20-dB-decay time itself. According to that definition, your reverberation times would need to be multiplied by six. Hence, your large room had a reverberation time of about 2.6 s, similar to a small cathedral.

We reported the RT_10_ values in our original manuscript because the time frame of RT_10_ (<600ms) is similar to that of the reverberation adaptation we have observed (over a timeframe of 200ms). However, the Reviewer makes a fair point about RT_60_ values being more comparable to other literature. We have edited the plot in question to now report both the RT_60_ (left y-axis) and RT_10_ (right y-axis) for the rooms we tested. We also now discuss our results primarily in terms of RT_60_ rather than RT_10_. As these values are just scaled versions of one another, the correlations remain unchanged.

Lines 373-382: “The decay rate can be measured as the reverberation time RT_60_, which is the time necessary for the sound level to decay by 60dB relative to an initial sound impulse (similarly, RT_10_ is the time necessary for a decay by 10dB). The frequency-dependent RT_60_ and RT_10_ values for our small and large rooms are plotted in Figure 7C. The impulse responses of both rooms exhibited a decrease in RT_60_ values as a function of frequency (Pearson's correlation, small room: r = -0.82, p = 1.1 x 10^-10^, large room: r = -0.91, p = 8.0 x 10^-10^). This faster decay for higher frequencies can also be observed in the spectrograms of the impulse responses (Figure 7—figure supplement 1). Therefore, the frequency-dependent delay in the inhibitory components of our dereverberation model and cortical STRFs paralleled the RT_60_ frequency profile of the virtual rooms in which the sounds were presented.”

We also added the RT_60_ values to our descriptions of the room sizes at the beginning of our Results section.

Lines 86-88: “The dimensions of the smaller room made it less reverberant (reverberation time: RT_10_ = 130ms, RT_60_ = 0.78s) than the larger room RT_10_ = 430ms, RT_60_ = 2.6s.”

And throughout our methods section. For example:

Lines 642-643: “The reverberation time RT_60_ is the time necessary for the sound level to decay by 60dB relative to an initial sound impulse, while RT_10_ is the time for the sound level to decay by 10dB.”

Reviewer #3 (Recommendations for the authors):This is a very interesting study that tests how the auditory system adapts to reverberant acoustic scenes. The paper is written well, and the results are overall compelling. I have a few comments that hopefully can strengthen the claims of the study.1. It is important to note that dereverberation requires highly nonlinear acoustic processing. The dereverberation algorithms that have been used in speech processing typically try to mask the spectrogram as a method to recover the "direct path" and remove the "reflections". While the resulting "high-pass" filter found in the linear filtering attempts to approximate this nonlinear operation, it is not very effective when used in realistic conditions. While the linear modeling framework here allows the authors to perform a straightforward comparison with auditory receptive fields, this limitation should be noted so the readers are aware of the true difficulty of this task and hence, unexplained mechanisms that remain to be found.

See response to Essential revision 3 above.

2. A big issue which the authors are well aware of is the differences between the stimuli used to find the receptive field in different reverberant scenarios. While the authors do a good job to show that the differences are not merely due to the statistics of the stimuli, particularly by showing a different response to the probe sound, they cannot claim that "all" of the observed changes are due to adaptation. It is likely reflecting a mix of both, some due to the change in the stimulus correlation and some due to adaptation. Currently, this inherent limitation is not acknowledged.

See response to Essential revision 5 above.

3. Related to 2, I think the control experiments that were done to show the changes in the response to the probe sound in different reverberant conditions are worthy of inclusion in the main figures. Perhaps a summary of that can be added to Figure 3, as this is a very important result, without which the observed changes are not very compelling.

We have moved these into the main manuscript figures. See response to Essential Revision 2 above.

4. On the same point, while it is an important observation that the responses to the probe stimulus (non-reverberant) are different in different reverberation contexts, a complementary observation that is missing is the similarity of the cortical responses to varying degrees of reverberation. In other words, if the STRFs change as suggested to produce a less variable response to reverberation, then the responses to the same sound with different reverb should stay constant and similar to the anechoic sound. While invariant cortical responses have been shown in other studies (including ferrets), it will strengthen this study if they can also confirm the presence of that effect which is the subject of this study.

See response to Essential revision 1 above.

5. Regarding the discussion of the feedforward/feedback nature of the adaptation to changing background statistics, Khalighinejad et al. also showed that the suppression of background noise is the same when the subject is actively performing speech-in-noise perception and when the subject is distracted by a visual task (Figure 5). Perhaps this observation can strengthen the argument regarding the anesthetized/awake conditions and the nature of the adaptation (lines 447-451).

See response to Essential revision 6 above.